# A Near-Optimal Best-of-Both-Worlds Algorithm for Online Learning with Feedback Graphs

**Chloé Rouyer**
Dept. of Computer Science
University of Copenhagen, Denmark
chloe@di.ku.dk

**Dirk van der Hoeven**
Dept. of Computer Science
Università degli Studi di Milano, Italy
dirk@dirkvanderhoeven.com

**Nicolò Cesa-Bianchi**
Dept. of Computer Science
Università degli Studi di Milano, Italy
nicolo.cesa-bianchi@unimi.it

**Yevgeny Seldin**
Dept. of Computer Science
University of Copenhagen, Denmark
seldin@di.ku.dk

## Abstract

We consider online learning with feedback graphs, a sequential decision-making framework where the learner's feedback is determined by a directed graph over the action set. We present a computationally efficient algorithm for learning in this framework that simultaneously achieves near-optimal regret bounds in both stochastic and adversarial environments. The bound against oblivious adversaries is $\tilde{O}(\sqrt{\alpha T})$, where $T$ is the time horizon and $\alpha$ is the independence number of the feedback graph. The bound against stochastic environments is $O\big((\ln T)^2 \max_{S \in \mathcal{I}(G)} \sum_{i \in S} \Delta_i^{-1}\big)$ where $\mathcal{I}(G)$ is the family of all independent sets in a suitably defined undirected version of the graph and $\Delta_i$ are the suboptimality gaps. The algorithm combines ideas from the EXP3++ algorithm for stochastic and adversarial bandits and the EXP3.G algorithm for feedback graphs with a novel exploration scheme. The scheme, which exploits the structure of the graph to reduce exploration, is key to obtain best-of-both-worlds guarantees with feedback graphs. We also extend our algorithm and results to a setting where the feedback graphs are allowed to change over time.

## 1 Introduction

Online learning is a general framework for studying sequential decision-making in unknown environments (see, for example, (Cesa-Bianchi and Lugosi, 2006; Bubeck and Cesa-Bianchi, 2012; Orabona, 2019)). We consider a setting where, at each round, the player chooses an action (a.k.a. arm) from a fixed set of $K$ actions and incurs the loss associated with the chosen action. The performance of the learner is quantified in terms of regret, which is the difference between the total loss incurred by the learner over the duration of the game, and the smallest cumulative loss obtained by a player that would only ever play the same action throughout the game.

The smallest achievable regret is determined by a number of parameters. One of these parameters is the amount of feedback that the learner receives at each round. There is a whole spectrum of problems, characterized by the amount of feedback received by the learner. At the one extreme of this spectrum is the bandit setting, where the learner only observes the loss of the action taken. At the other extreme is the full information setting, where the learner observes the full loss vector at the end of each round, irrespective of the action played.

36th Conference on Neural Information Processing Systems (NeurIPS 2022).

There are two common ways to interpolate between full information and bandit feedback. One is to allow the learner to make a limited number of additional observations without restricting how the additional observations are selected. Then no additional observations correspond to the bandit setting and $K - 1$ additional observations correspond to the full information setting. This way of interpolation was proposed by Seldin et al. (2014) in two variants, "prediction with limited advice" and "multiarmed bandits with paid observations". It was also studied by Thune and Seldin (2018).

The second way of interpolation, which we focus on in this paper, is via feedback graphs (Alon et al., 2017). In this setting observations of the learner are governed by a feedback graph on the actions. When an action is played, the learner observes the losses of all of its neighbors in the feedback graph. A complete graph corresponds to the full information setting, whereas a graph containing only self-loops corresponds to the bandit setting. This setting has multiple variants, depending on whether the graph is directed or undirected, observed or unobserved, static or dynamic.

Another important parameter characterizing online learning problems is the type of environment. The two primary types that we focus on are stochastic and adversarial environments. In stochastic environments each action is associated with a fixed, but unknown distribution, and in each round the loss of each action is sampled independently from the corresponding distribution. In adversarial environments the loss sequence is chosen arbitrarily. We consider oblivious adversarial environments, where the loss sequences are chosen independently of the actions taken by the learner.

For a long time stochastic and adversarial environments where studied separately, but in practice the exact nature of environment is rarely known. In recent years this has led to a growing interest in "best-of-both-worlds" algorithms that are robust against adversarial loss sequences and, at the same time, provide tighter regret guarantees in the stochastic regime. Most work has focused on the bandit setting (Bubeck and Slivkins, 2012; Seldin and Slivkins, 2014; Auer and Chiang, 2016; Seldin and Lugosi, 2017; Wei and Luo, 2018), where the Tsallis-INF algorithm proposed by Zimmert and Seldin (2019, 2021) was shown to achieve the optimal regret rates in both stochastic and adversarial regimes, as well as a number of intermediate regimes. The analysis was further improved by Masoudian and Seldin (2021) and Ito (2021). In the full information setting Mourtada and Gaïffas (2019) have shown that the well-known Hedge algorithm originally designed for the adversarial setting (Littlestone and Warmuth, 1994) also achieves the optimal stochastic regret. Best-of-both-world results also spilled over to other domains, including additional approaches to full information games and online convex optimization (Koolen et al., 2016; Van Erven et al., 2021; Negrea et al., 2021), decoupled exploration and exploitation (Rouyer and Seldin, 2020), combinatorial bandits (Zimmert et al., 2019), bandits with switching costs (Rouyer et al., 2021), MDPs (Jin and Luo, 2020; Jin et al., 2021), and linear bandits (Lee et al., 2021).

In the context of online learning with feedback graphs the only best-of-both-worlds result known to us is by Erez and Koren (2021) for undirected graphs. They present an intricate algorithm based on the Follow The Regularized Leader (FTRL) framework with a regularization function that is a product of the Tsallis and Shannon entropies. The algorithm simultaneously enjoys an $O\big(\sqrt{\chi T}\big(\ln(KT)\big)^2\big)$ pseudo-regret bound in the adversarial regime and an $O\big(\big(\ln(KT)\big)^4 \sum_k \frac{\ln T}{\Delta_k}\big)$ pseudo-regret bound in the stochastic regime, where $T$ is the number of prediction rounds, $\chi$ is the clique covering number of the undirected feedback graph, and the summation in the second bound is on the smallest non-zero gap within each clique.

It is tempting to apply an FTRL-based algorithm with Tsallis entropy regularization to online learning with feedback graphs, since Tsallis entropy with power $a = 1/2$ leads to the optimal Tsallis-INF algorithm for the bandit setting (Zimmert and Seldin, 2021) and Tsallis entropy with power $a = 1$ leads to the Hedge algorithm, which is optimal in the full information setting. However, as also noted by Erez and Koren (2021), extension of the analysis to online learning with feedback graphs when the power $a \in (1/2, 1)$ is not straightforward and, so far, there was no success in this direction. Furthermore, at the moment it is unclear whether it is possible to derive bounds that take further advantage of the graph structure and depend on the independence number of the graph when $a < 1$.

**Our contribution.** We significantly extend and improve on the bounds of Erez and Koren (2021). Our results hold for directed graphs (with self-loops), depend on the independence number of the graph, have a better dependence on $T$ in the stochastic regime, and extend to time-varying feedback graphs. Our approach takes advantage of the common structure shared by two exponential weights algorithms: EXP3.G (Alon et al., 2015) and EXP3++ (Seldin and Slivkins, 2014; Seldin and Lugosi,

2017), to obtain near-optimal best-of-both worlds guarantees. By using similar ideas as in the proof of the regret bound of EXP3.G, the proposed algorithm adapts to the independence number of the graph. We derive an $\min\left\{O(\sqrt{\tilde{\alpha}T\ln K}), O\left(\sqrt{\ln K}\sqrt{\ln(KT)}\sqrt{\alpha T}\right)\right\}$ pseudo-regret bound against adversarial sequences of losses, where $\alpha$ is the independence number of the graph and $\tilde{\alpha}$ is its strong independence number, which is a graph dependent quantity smaller than the clique covering number. For undirected graphs, independence number and strong independence number are equal and the result matches the best known lower bound $\Omega(\sqrt{\alpha T})$ within logarithmic factors (Alon et al., 2017). In the stochastic setting we use the idea of injected exploration from EXP3++ to estimate the suboptimality gaps of each arm. By introducing a novel dynamic exploration set and an appropriate exploration rate, we derive an almost optimal regret bound in the stochastic setting. Along the way, we also improve the regret bound of EXP3++ in the stochastic bandit setting. Our exploration set is constructed by sorting the arms by ascending gap estimates, and then adding a new arm to the exploration set if the arm cannot be observed by playing another arm previously added to the set. If we play each arm $i$ in the exploration set at a rate $1/\hat{\Delta}_i^2$, where $\hat{\Delta}_i$ is the gap estimate, then all arms $j$ in the graph are observed with probability at least $1/\hat{\Delta}_j^2$.

To present our main result we introduce some notations. Let $G = (V, E)$ be a directed feedback graph with independence number $\alpha$ (where the independence number is computed on $G$ ignoring edge directions). We define a strongly independent set on $G$ as an independent set on the subgraph $G' = (V, E')$, where $(i, j) \in E'$ if and only if $(i, j) \in E$ and $(j, i) \in E$. We use $\tilde{\alpha}$ to denote the strong independence number of $G$, and $\mathcal{I}(G)$ to denote the collection of all the strongly independent sets in $G$. We note that $\alpha = \tilde{\alpha}$ for undirected graphs and $\alpha \leq \tilde{\alpha}$ for directed graphs. Now we can present an informal statement of our main result.

**Theorem 1** (Informal). *Given a directed feedback graph $G = (V, E)$ with independence number $\alpha$ and strong independence number $\tilde{\alpha}$, there exists an algorithm (Algorithm 1) whose pseudo-regret can simultaneously be bounded by $\min\left\{O(\sqrt{\tilde{\alpha}T\ln K}), O\left(\sqrt{\ln K}\sqrt{\ln(KT)}\sqrt{\alpha T}\right)\right\}$ against adversarial loss sequences and by $O\left((\ln T)^2 \max_{S \in \mathcal{I}(G)} \sum_{i \in S} \Delta_i^{-1}\right)$ against stochastic loss sequences.*

We emphasize that Algorithm 1 requires neither prior knowledge of the type of the environment (adversarial or stochastic), nor the time horizon.

## 1.1 Additional Related Work

The study of bandits with feedback graphs was initiated by Mannor and Shamir (2011) in the adversarial regime and by Caron et al. (2012) in the stochastic regime. In the adversarial regime, the optimal regret rates for arbitrary directed graphs were characterized (up to log factors) by Alon et al. (2015). They showed an $\Omega(T)$ lower bound for graphs that have non-observable nodes (i.e., with an empty in-neighborhood). For graphs with observable nodes, they derived pseudo-regret bounds of order $O\left(\sqrt{\alpha T}\log(KT)\right)$ when all nodes are strongly observable (i.e., they have a self-loop or their in-neighborhood contains all of the other nodes) and of order $O\left((\delta \ln K)^{1/3}T^{2/3}\right)$ for weakly observable graphs (where each non-strongly observable node is in the out-neighborhood of some observable node). Here $\alpha$ is the independence number of the graph and $\delta$ is the dominating number of the weakly observable portion of the graph. Van der Hoeven et al. (2021) derived results for the multiclass classification with feedback graphs setting. The setting where the graph can adversarially change over time has been studied by Alon et al. (2017) in the case of directed graphs with self-loops. For learners that are allowed to observe the feedback graph at the beginning of each round, they achieved a bound of $O\left(\ln K \sqrt{\ln(KT)\sum_{t=1}^{T}\alpha_t}\right)$, where $\alpha_t$ is the independence number of the graph at time $t$. For the case of undirected graphs, they proved a refined bound $O\left(\sqrt{\ln K \sum_{t=1}^{T}\alpha_t}\right)$ that holds even when the learner can only observe the graph at the end of each round. Note that, as shown by Cohen et al. (2016), in order to take advantage of the graph structure in the adversarial regime, it not sufficient to observe the neighborhood of the played action at the end of each round.

In the stochastic regime, Buccapatnam et al. (2014, 2017) considered a fixed, possibly directed, feedback graph. They derived an asymptotic lower bound showing that the regret scales as $\Omega(c^* \ln T)$, where $c^*$—which is related to the domination number of the graph—is the solution to a linear program expressing the trade-off between the loss incurred from playing an action and the observations that

can be gathered from playing that action. They proposed an algorithm that can achieve a matching $O(c^* \ln T + Kd)$ pseudo-regret bound, where $d$ is the maximum degree in the feedback graph. In the case of graphs that change over time, Cohen et al. (2016) derived an $O(\sum_{i \in S} (\ln T)/\Delta_i)$ bound, where $S$ is a set containing an order of $\alpha$ arms (up to log factors), and $\alpha$ is an upper bound on the independence number of the graphs in the sequence. They achieved this result without requiring to observe the graphs fully, and having only access to the neighbourhood of the arm played at the end of the round. Both of these approaches are based on arm elimination algorithms, which—by construction—are not suitable for best-of-both worlds guarantees. The proof strategy of Cohen et al. (2016) was adapted by Lykouris et al. (2020) to provide refined bounds for both UCB-N and Thompson Sampling-N, which are variants of UCB1 (Auer et al., 2002) and Thompson Sampling (Thompson, 1933). In both cases, Lykouris et al. (2020) considered undirected feedback graphs and obtained pseudo-regret bounds that scale as $O\big(\max_{\text{Ind} \in \mathcal{I}(G)} \sum_{i \in \text{Ind}} \ln(KT)(\ln T)/\Delta_i\big)$, where $\mathcal{I}(G)$ is the collection of all the independence sets of the graph.

Concurrently to our work, several other papers in online learning with feedback graphs have appeared. Ito et al. (2022) derive an algorithm with nearly optimal regret bounds in both the stochastic and adversarial setting. While their results are more general than ours (they do not require self-loops in the feedback graph), their regret bounds in the stochastic regime are worse that ours, of order $\frac{\ln(T)^3}{\Delta_{\min}}$, where $\Delta_{\min}$ is the minimum suboptimality gap. Similarly to Erez and Koren (2021), the algorithm of Ito et al. (2022) is based on the FTRL framework. They use the entropic regularization, which makes their algorithm equivalent to EXP3 (Auer et al., 2002). Moreover, Ito et al. (2022) rely on the self-bounding technique of Zimmert et al. (2019) together with an intricate tuning to simultaneously obtain regret bounds in the stochastic regime and the adversarial regime, as well as in intermediate ones. In the stochastic regime, Marinov et al. (2022) provide an improved characterization of the difficulty of online learning with feedback graphs in both finite-time and asymptotic cases. Finally, Esposito et al. (2022) study the more general model of stochastic feedback graphs.

## 2 Problem Setting and Definitions

**Problem Setting** We consider a sequential decision-making game, where in each round $t = 1, 2, \ldots$, the learner repeatedly plays an action $I_t \in V$, where $|V| = K$, receives a feedback based on a feedback graph $G = (V, E)$, and suffers a loss $\ell_{t, I_t}$. We consider directed feedback graphs with self-loops, meaning that $(i, i) \in E$ for each vertex $i \in V$. The feedback received by the learner at the end of round $t$ is $\big\{(i, \ell_{t,i}) : i \in N^{\text{out}}(I_t)\big\}$, where $N^{\text{out}}(i) = \{j \in V : (i, j) \in E\}$ is the out-neighbourhood of $i$. Similarly, we define $N^{\text{in}}(i) = \{j \in V : (j, i) \in E\}$ to be the in-neighborhood of $i$. For each arm $i \in V$, $\ell_{t,i} \in [0, 1]$ for $t \geq 1$. In the adversarial regime the losses are generated arbitrarily by an oblivious adversary. In the stochastic regime they are independently drawn from a fixed but unknown distribution with expectation $\mathbb{E}[\ell_{1,i}]$. The performance of the learner is measured in terms of the pseudo-regret:

$$\mathcal{R}_T = \mathbb{E}\left[\sum_{t=1}^T \ell_{t, I_t}\right] - \min_{i \in V} \mathbb{E}\left[\sum_{t=1}^T \ell_{t,i}\right].$$

In the stochastic regime, we define the best arm $i^*$ as the arm with the smallest expected loss, i.e. $i^* = \text{argmin}_{i \in V} \mathbb{E}[\ell_{1,i}]$. The pseudo-regret can then be expressed in terms of the suboptimality gaps $\Delta_i = \mathbb{E}[\ell_{1,i} - \ell_{1,i^*}]$,

$$\mathcal{R}_T = \sum_{t=1}^T \sum_{i \in V} \mathbb{E}[p_{t,i}] \Delta_i, \tag{1}$$

where $p_{t,i}$ is the probability that the learner plays action $i$ at round $t$. We define the smallest suboptimality gap $\Delta_{\min} = \min_{i: \Delta_i > 0} \{\Delta_i\}$, and for all $i$, we define $\bar{\Delta}_i = \max\{\Delta_{\min}, \Delta_i\}$, so that $\bar{\Delta}_{i^*} = \Delta_{\min}$. We use $\mathbb{E}_t$ to express expectation conditioned on all randomness up to round $t$.

**Properties of Graphs** Recall that a dominating set in $G$ is a subset $D \subseteq V$, such that for all $i \in V$ there exists $j \in D$, such that $(j, i) \in E$. An independent set in $G$ is a subset $S \subseteq V$, such that for all $i, j \in S$, $(i, j) \notin E$ and $(j, i) \notin E$. We define the independence number $\alpha(G)$ as the size of the largest independent set in the graph $G$. For clarity, we restate below here the definition of the strong independence number which was already mentioned in the introduction.

**Algorithm 1:** EXP3.G++

**Input:** Feedback graph $G = (V, E)$,
Learning rates $\eta_1 \geq \eta_2 \geq \cdots > 0$; exploration rates $\varepsilon_{t,i}$ for $i \in V$, see Equation (2)
**Initialize:** $\tilde{L}_0 = \mathbf{0}_K$, $\hat{L}_0 = \mathbf{0}_K$ and $O_0 = \mathbf{0}_K$. Play each arm once to initialize $\hat{L}$ and $O$
**for** $t = K + 1, K + 2, \ldots$ **do**

$$\forall i \in V : \text{UCB}_{t,i} = \min \left\{ 1, \frac{\hat{L}_{t-1,i}}{O_{t-1,i}} + \sqrt{\frac{\gamma \ln \left( t K^{1/\gamma} \right)}{2 O_{t-1,i}}} \right\}$$

$$\forall i \in V : \text{LCB}_{t,i} = \max \left\{ 0, \frac{\hat{L}_{t-1,i}}{O_{t-1,i}} - \sqrt{\frac{\gamma \ln \left( t K^{1/\gamma} \right)}{2 O_{t-1,i}}} \right\}$$

$\forall i \in V : \hat{\Delta}_{t,i} = \max \left\{ 0, \text{LCB}_{t,i} - \min_j \text{UCB}_{t,j} \right\}$
$\forall i \in V :$ update $\varepsilon_{t,i}$ based on the gap estimates $\hat{\Delta}_t$

$$\forall i \in V : q_{t,i} = \frac{\exp(-\eta_t \tilde{L}_{t,i})}{\sum_{i \in V} \exp(-\eta_t \tilde{L}_{t,j})}, \; p_{t,i} = \left( 1 - \sum_{j \in V} \varepsilon_{t,j} \right) q_{t,i} + \varepsilon_{t,i}$$

Sample $I_t \sim p_t$ and play it
Observe $\{ (j, \ell_{t,j}) : j \in N^{\text{out}}(I_t) \}$ and suffer $\ell_{t,I_t}$.
$\forall i \in V : \tilde{\ell}_{t,i} = \dfrac{\ell_{t,i} \mathbb{1} \left[ i \in N^{\text{out}}(I_t) \right]}{P_{t,i}}$, where $P_{t,i} = \sum_{j \in N^{\text{in}}(i)} p_{t,j}$
$\forall i \in V : \quad \tilde{L}_{t,i} = \tilde{L}_{t-1,i} + \tilde{\ell}_{t,i}$
$\forall i \in V : \hat{L}_{t,i} = \hat{L}_{t-1,i} + \ell_{t,i} \mathbb{1} \left[ i \in N^{\text{out}}(I_t) \right]$ and $O_{t,i} = O_{t-1,i} + \mathbb{1} \left[ i \in N^{\text{out}}(I_t) \right]$
**end for**

**Definition 1.** *Let $G = (V, E)$ be a directed graph. We define a strongly independent set on $G$ as an independent set on the subgraph $G' = (V, E')$, where $(i, j) \in E'$ if and only if $(i, j) \in E$ and $(j, i) \in E$. Furthermore, we define $\tilde{\alpha}(G)$ as the independence number of the subgraph $G'$.*

We use $\mathcal{I}(G)$ to denote a collection of all the strongly independent sets in $G$. We note that $\alpha = \tilde{\alpha}$ for undirected graphs and $\alpha \leq \tilde{\alpha}$ for directed graphs.

## 3 Algorithm

We present the EXP3.G++ algorithm (Algorithm 1), which is a combination of the EXP3.G algorithm of Alon et al. (2015) and the EXP3++ algorithm of Seldin and Lugosi (2017) with a novel exploration scheme described in Algorithm 2. This scheme ensures that the additional feedback the learner obtains (relative to the bandit setting) is used nearly optimally.

To understand the motivation behind the novel exploration scheme, note that in the stochastic setting EXP3.G++ needs to ensure that the loss of each arm is observed sufficiently often. However, if we would play each arm too often, the regret would scale with the number of arms, rather than with the independence number or some other graph-theoretic quantity. To avoid that, we exploit the central property of feedback graphs: since we can gather information on certain arms by playing adjacent arms in the graph, we can restrict exploration to a subset of nodes and yet obtain sufficient information on *all* the arms. We exploit this observation, to design a strategy for selecting an exploration set $S_t$ at each round $t$. $S_t$ is defined in terms of estimated suboptimality gaps $\hat{\Delta}_{t,i}$, which are maintained by EXP3.G++. Crucially, the exploration set ensures that, with high probability, the empirical gaps are reliable estimates of the true suboptimality gaps $\Delta_i$. In turn, this ensures that we observe the loss of each arm sufficiently often.

The construction of the exploration set $S_t$ is detailed in Algorithm 2, which is used by EXP3.G++ to update the exploration rates $\varepsilon_{t,i}$ according to Equation (2). Algorithm 2 starts by sorting the arms according to their gap estimates in ascending order. The exploration set is then greedily constructed by sequentially selecting the next arm with the smallest $\hat{\Delta}_{t,i}$, and discarding all the arms in the out-neighborhood of that arm. The exploration set can be constructed in $O(K^3)$ time, but note that

**Algorithm 2:** Exploration Set Construction

---

**Input:** $K$ arms with associated gaps: $\Delta_1, \Delta_2, \dots$
**Initialize:** Exploration set $S = \emptyset$.
Let $\Lambda$ be the list of arms sorted in ascending order of their associated gaps.
**for** $i \in \Lambda$ **do**
    Add $i$ to $S$
    **for** $j \in N^{\mathrm{out}}(i)$ **do**
        remove $j$ from $\Lambda$
    **end for**
**end for**
**Output:** $S$

---

we only need to recompute it only when the order of the estimated suboptimality gaps changes. The exploration set $S_t$ has several useful properties, as shown in Proposition 1 below.

**Proposition 1.** *Let $G = (V, E)$ be a directed feedback graph on $K$ arms with self-loops, and let $\hat{\Delta}_1, \dots, \hat{\Delta}_K$ be a sequence of suboptimality gaps estimates. Let $S$ be the exploration set constructed by Algorithm 2 based on the sequence of suboptimality gaps. Then $S$ is a dominating set of $G$ with the following property: for all $i \in V$ there exists $j \in S$, such that $i \in N^{\mathrm{out}}(j)$ and $\hat{\Delta}_j \leq \hat{\Delta}_i$. Furthermore, $S$ is also a strongly independent set of $G$.*

*Proof.* Let $S$ be the output of Algorithm 2. Since $G$ contains self-loops, if $i \in S$, then $i \in N^{\mathrm{out}}(i)$ and $\hat{\Delta}_i \leq \hat{\Delta}_i$. If $i \notin S$, then $i$ was removed from $\Lambda$ because $i \in N^{\mathrm{out}}(j)$ for some $j$ that, in a previous iteration, was added to $S$. Since $j$ was considered before $i$, we must have $\hat{\Delta}_j \leq \hat{\Delta}_i$. Now, for all $i, j \in S$, we know by construction that $j \notin N^{\mathrm{out}}(i)$. Thus $(i, j)$ is not a directed edge in $G$, and so $S$ is a strongly independent set in $G$. $\qquad\square$

We define the exploration rates at round $t$ in terms of the exploration set $S_t$, which is constructed using the aforementioned procedure. For all arms $i$ in $V$,

$$\varepsilon_{t,i} = \min\left\{ \frac{1}{2K}, \frac{1}{2}\sqrt{\frac{\lambda \ln K}{tK^2}}, \xi_{t,i} \right\}, \tag{2}$$

for some constant $\lambda \in [1, K]$ and where $\xi_{t,i}$ depends on whether $i \in S_t$ or not:

$$\xi_{t,i} = \begin{cases} (\beta \ln t)/(t\hat{\Delta}_{t,i}^2), & \text{if } i \in S_t, \\ 4/t^2, & \text{otherwise,} \end{cases} \tag{3}$$

where $\beta > 0$ is a constant. The role of $\xi_{t,i}$ changes depending on whether we are in an adversarial or stochastic environment. In an adversarial environment, we use $4/t^2 \leq \xi_{t,i}$ to ensure that we sample each arm with a small positive probability, which is essential to bound the second-order term in the regret bound in terms of the independence number. Note that $\varepsilon_{t,i} \leq \frac{1}{2}\sqrt{\frac{\lambda \ln K}{tK^2}}$, so choosing $\lambda = \alpha$ ensures that the cost of exploration is bounded by $\tilde{O}(\sqrt{\alpha T})$. In the stochastic environment, the construction of the exploration set and the choice of $\xi_{t,i}$ ensure that, at each round $t$, each $i \in V$ is observed with probability at least $(\beta \ln t)/(t\hat{\Delta}_{t,i}^2)$, independently of whether $i$ is in the exploration set at round $t$.

Formally, our procedure ensures that we can lower bound the probability with which any arm is observed. In the algorithm we use $P_{t,i} = \mathbb{P}\left[i \in N^{\mathrm{out}}(I_t)\right]$ to denote the probability that arm $i$ is observed at round $t$. We can lower bound this quantity by only considering the minimum rate at which each arm is observed according to the exploration rate $\varepsilon_{t,i}$ and our construction of the exploration sets. We use $o_{t,i}$ to denote that quantity, and we have for all $t$ and $i$,

$$P_{t,i} \geq o_{t,i} = \min\left\{ \frac{1}{2K}, \frac{1}{2}\sqrt{\frac{\lambda \ln K}{tK^2}}, \frac{\beta \ln t}{t\hat{\Delta}_{t,i}^2} \right\}. \tag{4}$$

The definition of $o_{t,i}$ uses that $S_t$ is a dominating set. The difference between $\varepsilon_{t,i}$ and $o_{t,i}$ is key to take advantage of the graph structure. First, we need to lower bound $o_{t,i}$ to ensure that enough observations (counted by $O_{t,i}$ in Algorithm 1) are made for each arm, such that our gap estimates are reliable. Simultaneously, we upper bound $\varepsilon_{t,i}$ to ensure that the extra exploration is not too costly. Here we benefit from the fact that $S_t$ is a strongly independent set on $G$.

We ensure that all arms get sufficiently many observations and derive the following concentration bounds on the gap estimates $\hat{\Delta}_{t,i}$ computed by Algorithm 1. Concentration of the gap estimates around the true gaps is crucial for bounding the regret in the stochastic setting.

**Lemma 1.** *If Algorithm 1 is run with parameters $\gamma \geq 3$, $\beta \geq 64(\gamma + 1) \geq 256$, and exploration rates $\varepsilon_{t,i}$, such that for all $t \geq 1$ and $i \in V$, $P_{t,i}$ satisfies equation (4) for some $\lambda \in [1, K]$, then for all $i \in V$ and $t \geq 1$,*

$$\mathbb{P}\left[\hat{\Delta}_{t,i} \geq \overline{\Delta}_i\right] \leq \frac{1}{Kt^{\gamma-1}}.$$

*Furthermore, for any arm $i$ with $\Delta_i > 0$ let $t_{\min}(i) := \max\left\{t \geq 0 : \frac{1}{2}\sqrt{\frac{\lambda \ln K}{tK^2}} \leq \frac{\beta \ln t}{t\Delta_i^2}\right\}$. Then for any arm $i$ with $\Delta_i > 0$ and $t \geq t_{\min}(i)$,*

$$\mathbb{P}\left[\hat{\Delta}_{t,i} \leq \frac{1}{2}\Delta_i\right] \leq \left(\frac{\ln t}{t\Delta_i^2}\right)^{\gamma-2} + \frac{2}{Kt^{\gamma-1}} + 2\left(\frac{1}{t}\right)^{\frac{\beta}{10}}. \tag{5}$$

A proof of the lemma is provided in Appendix C.

We run the algorithm with $\gamma = 4$ and $\beta = 64(\gamma + 1) = 320$ which is a different parameterization from the EXP3++ algorithm (Seldin and Lugosi, 2017), which uses $\gamma = 3$ and $\beta = 256$. Picking a larger value of $\gamma$ means that the confidence intervals are slightly larger, which allows us to obtain a better dependency on the suboptimality gaps.

Indeed, under the same assumptions as in Lemma 1, if $\gamma = 4$ and $t \geq t_{\min}(i)$, we have that

$$\frac{(\ln t)^2}{t} \leq \frac{\lambda \Delta_i^4 \ln K}{4K^2\beta^2}, \text{ implying } \left(\frac{\ln t}{t\Delta_i^2}\right)^2 = \frac{(\ln t)^2}{t^2\Delta_i^4} = \frac{(\ln t)^2}{t}\frac{1}{t\Delta_i^4} \leq \frac{\lambda \Delta_i^4 \ln K}{4K^2\beta^2}\frac{1}{t\Delta_i^4} = \frac{1}{t}\frac{\lambda \ln K}{4K^2\beta^2}.$$

## 4  Adversarial Analysis

Our result for the adversarial regime generalizes the analysis of both Alon et al. (2017) and Alon et al. (2015) as we derive a bound that depends on the both the independence number and the strong independence number simultaneously. In order to do so, we define the quantity:

$$\theta_t := \sum_{i \in V} \frac{p_{t,i}}{P_{t,i}}, \tag{6}$$

which is the sum of the ratios of the probability of playing an arm to the probability of observing its loss. Bounding this sum of ratios is key to obtain a dependency on graph quantities, and Alon et al. (2017) and Alon et al. (2015) respectively bound equation (6) in terms of the strong independence number (Lemma 8) and the independence number (Lemma 7) at the cost of a logarithmic factor. By defining the learning rate in terms of $\theta$, it is possible to obtain both bounds simultaneously.

**Theorem 2.** *Assume that Algorithm 1 is run with a directed feedback graph $G = (V, E)$, with learning rate $\eta_t = \sqrt{\frac{\ln K}{2\sum_{s=K}^{t-1}\theta_s}}$ and the exploration rate defined in (2)–(3) with $\gamma = 4$, and $\beta = 320$. For any $\lambda \in [1, \min(\tilde{\alpha}, \alpha \ln T)]$, the pseudo-regret against any oblivious loss sequence satisfies*

$$\mathcal{R}_T \leq \min\left\{4\sqrt{\tilde{\alpha}T\ln K}, 9\sqrt{\ln K}\sqrt{\ln(KT)}\sqrt{\alpha T}\right\} + 2K,$$

*where $K = |V|$, $\alpha$ is the independence number of $G$ and $\tilde{\alpha}$ is its strong independence number.*

On undirected graphs, the first part of the bound is always smaller, and it matches the bound of Alon et al. (2017). This implies that in the adversarial regime we are not paying a price for the extra guarantees that we derive in the stochastic regime. On directed graphs, if the difference between $\alpha$ and $\tilde{\alpha}$ is large, the second half of the bound may be advantageous. Furthermore, we note that the extra

logarithmic factor is only of order $\sqrt{\ln(T)}$, which is a slight improvement on the $\ln T$ dependency of Alon et al. (2015).

We give a sketch of the proof here and defer the detailed proof to Appendix B.

**Proof sketch.** We separate the first $K$ rounds, in which the algorithm plays deterministically, from the remaining rounds, where we bound separately the contributions to the regret from the exponential weights and from the extra exploration. To bound the contribution of the extra exploration, we use that $\varepsilon_{t,i} \leq \frac{1}{2}\sqrt{\frac{\lambda \ln K}{tK^2}}$ for all $t$ and $i$, meaning that the extra exploration contributes at most $O(\sqrt{\lambda T \ln K})$ to the regret. For bounding the contribution of the exponential weights to the regret, we follow the standard analysis of EXP3 with time varying learning rate (Bubeck and Cesa-Bianchi, 2012). We bound the second order term by exploiting the fact that $p_{t,i}$ and $q_{t,i}$ are close to each other because $\varepsilon_{t,i} \leq \frac{1}{2K}$ for all $t$ and $i$. This allows us to bound the second order term in terms of $\theta_t$, which simplifies with the learning rate. This quantity can be bounded by the strong independence number of the graph (Lemma 10 Alon et al. (2017)) or the independence number of the graph at the cost of a logarithmic factor (Lemma 5 Alon et al. (2015)), which gives the two parts of the bound.

## 5 Stochastic Analysis

In the stochastic regime, tuning $\lambda$ affects the tightness of the bound. If the learner has knowledge of the independence and strong independence numbers but does not know the time horizon, picking $\lambda = \alpha$ is a safe choice to ensure that Theorem 2 holds.

Our result for the stochastic regime is given in the following theorem.

**Theorem 3.** *Let $G = (V, E)$ be a directed feedback graph with $K = |V|$ and independence number $\alpha$ and strong independence number $\tilde{\alpha}$. Under the same conditions as in Theorem 2 and choosing $\lambda = \alpha$, the pseudo-regret of Algorithm 1 against any stochastic stochastic loss sequence, satisfies:*

$$\mathcal{R}_T \leq \max_{\mathrm{Ind} \in \mathcal{I}(G)} \left\{ \sum_{i \in \mathrm{Ind} \,:\, \Delta_i > 0} \frac{4\beta \left(\ln T\right)^2}{\Delta_i} \right\} + 2\alpha \ln T$$
$$+ \sum_{i\,:\,\Delta_i>0} \frac{16K}{\Delta_i} + \frac{1020\beta K}{\Delta_{min}^2} \left( \ln\left( \frac{\beta K}{\Delta_{min}} \right) \right)^{3/2},$$

*where $\mathcal{I}(G)$ is the collection of all strongly independent subsets of $G$.*

We remark that the last two terms do not depend on $T$. Moreover, the leading coefficient of the term scaling with $(\ln T)^2$ sums over an independence set (as opposed to summing over the entire action set). The lower bound for this problem scales as $\Omega(c^* \ln T)$, where $c^*$ is a graph dependent quantity which takes the size of the suboptimality gaps into account (Buccapatnam et al., 2014). Compared to that, our result is suboptimal by a logarithmic factor and our dependency on the strong independence number of $G$ is weaker. The algorithms of Buccapatnam et al. (2014, 2017) and Cohen et al. (2016) almost match the lower bound, but their elimination based structure prevents them from being applicable in best-of-both-worlds settings. In the undirected case, we obtain the same dependence on $T$ and on the set of arms as the UCB-N algorithm analysed by Lykouris et al. (2020).

We provide a sketch of the proof here. The detailed version can be found in Appendix D.

**Proof sketch.** Let $t_{\min} = \max_{i:\Delta_i>0}\{t_{\min}(i)\} = \max\left\{ t \geq 0 : \frac{1}{2}\sqrt{\frac{\alpha \ln K}{tK^2}} \leq \frac{\beta \ln t}{t\Delta_{\min}^2} \right\}$. The pseudo-regret can be decomposed by treating the first $t_{\min}$ rounds like in the adversarial case, and by using a refined bound for the stochastic regime in the remaining rounds.

$$R_T = R_{t_{\min}} + \sum_{t=t_{\min}}^{T} \sum_{i\,:\,\Delta_i>0} \Delta_i \mathbb{E}\left[p_{t,i}\right] \leq R_{t_{\min}} + \sum_{t=t_{\min}}^{T} \sum_{i\,:\,\Delta_i>0} \Delta_i \big(\mathbb{E}\left[q_{t,i}\right] + \mathbb{E}\left[\varepsilon_{t,i}\right]\big). \quad (7)$$

Note that $t_{\min}$ is time independent: $t_{\min} = \frac{c}{\Delta_{\min}^4}\left(\ln\left(\frac{c}{\Delta_{\min}^4}\right)\right)^2$ for a positive constant $c$, therefore,

$$R_{t_{\min}} = C_0\sqrt{\alpha\, t_{\min}}\log\left(t_{\min}\right) = C_1 \frac{K}{\Delta_{\min}^2}\left(\ln\left(\frac{K}{\Delta_{\min}}\right)\right), \quad (8)$$

where the first equality follows from the second part of the bound presented in Theorem 2 and $C_0, C_1$ are universal constants. After the initial $t_{\min}$ rounds, enough observations on all arms have been gathered to ensure with high probability that the gap estimates of all arms are close to their true gaps, as stated in Lemma 1. These concentration inequalities allow us to show that the two following propositions hold.

**Proposition 2** (informal). *The contribution of the exponential weights to the pseudo-regret can be bounded as:*

$$\sum_{t=t_{\min}}^{T} \sum_{i\,:\,\Delta_i>0} \Delta_i \mathbb{E}\left[q_{t,i}\right] \leq C_2 \sum_{i\,:\,\Delta_i>0} \frac{K}{\Delta_i} + O\left(\alpha \ln T\right)$$

*for a universal constant $C_2$.*

**Proposition 3** (informal). *The contribution of the extra exploration to the pseudo-regret can be bounded as:*

$$\sum_{t=t_{\min}}^{T} \sum_{i\,:\,\Delta_i>0} \Delta_i \mathbb{E}\left[\varepsilon_{t,i}\right] = O\left(\max_{\text{Ind}\in\mathcal{I}(G)} \left\{\sum_{i\in\text{Ind}\,:\,\Delta_i>0} \frac{\ln^2 T}{\Delta_i}\right\} + \alpha \ln T\right).$$

Formal statements and proofs of the above propositions are in Appendix D. These propositions ensure that after $t_{\min}$ steps the exponential weights of all suboptimal arms $i$ are small, the extra exploration $\varepsilon_{t,i}$ achieves the correct rate, and that the sum of the probabilities that the suboptimality gap estimates fail in any of the rounds is of order $O(\alpha \ln T)$. Applying these propositions to Equation (7) finishes the proof.

Our approach to bound the pseudo-regret in the initial rounds differs from the one of Seldin and Lugosi (2017) as we take advantage of the adversarial bound in these rounds. (Mourtada and Gaïffas (2019) used a similar approach to derive best-of-both worlds guarantees for the Hedge algorithm.) This refinement improves upon the result of Seldin and Lugosi (2017) by replacing $\sum_{i:\Delta_i>0} \frac{1}{\Delta_i^3}$ with $\frac{1}{\Delta_{\min}^2}$ (numerical constants ignored) in the time-independent part of the bound.

For instances where the independence number and the strong independence number are close to each other, in particular in the case of undirected graphs, the analysis of the initial rounds can be improved by using the first part of Theorem 2, which depends on $\tilde{\alpha}$ rather than the second part, which depends on $\alpha$ when bounding the regret on the initial $t_{min}$ rounds.

**Corollary 1.** *Let $G = (V, E)$ be a directed feedback graph with $K = |V|$ and a strong independence number $\tilde{\alpha}$. Under the same conditions as in Theorem 2, the pseudo-regret of Algorithm 1 against any stochastic stochastic loss sequence, satisfies:*

$$\mathcal{R}_T \leq \max_{\text{Ind}\in\mathcal{I}(G)} \left\{\sum_{i\in\text{Ind}\,:\,\Delta_i>0} \frac{4\beta\left(\ln T\right)^2}{\Delta_i}\right\} + 2\tilde{\alpha}\ln T + \sum_{i\,:\,\Delta_i>0} \frac{16K}{\Delta_i} + \frac{161\beta K}{\Delta_{min}^2} \ln\left(\frac{\sqrt{\beta}K}{\Delta_{min}}\right).$$

## 6 Extension to Time Varying Feedback Graphs

The results presented in Theorem 3 and Corollary 1 assume that the learner has knowledge of the independence and strong independence numbers of the graph. Computing those numbers are NP-hard problems, which could lead to prohibitively large computation times. This is particularly true if one considers a natural extension of our results to the setting where the feedback graphs are allowed to change over time.

We consider a setting where an oblivious adversary chooses a feedback graph at each round and the algorithm observes the graph at the beginning of the round. In the stochastic regime, the knowledge of the full feedback graph is required at the beginning of the round in order to construct the exploration set.

As we do not know the independence numbers ahead of time, we tune the exploration rates defined in equation (2) with $\lambda = 1$ to ensure that the exploration is never too large. This exploration rate allows us to apply Lemma 1 with $\lambda = 1$, and derive the following result.

**Theorem 4.** *Assume that Algorithm 1 is run on a sequence of arbitrarily generated feedback graphs $G_1, G_2, \ldots$ with learning rate $\eta_t = \sqrt{\frac{\ln K}{2 \sum_{s=K}^{t-1} \theta_s}}$ and exploration rates defined in (2) and (3) with $\lambda = 1$, $\gamma = 4$ and $\beta = 320$. Then the pseudo-regret against any oblivious loss sequence satisfies*

$$\mathcal{R}_T \leq \min \left\{ 4\sqrt{\sum_{t=1}^{T} \tilde{\alpha}_t \ln K}, \ 9\sqrt{\ln K}\sqrt{\ln(KT)}\sqrt{\sum_{t=1}^{T} \alpha_t} \right\} + 2K,$$

*where for all $t \geq 1$, $\alpha_t$ and $\tilde{\alpha}_t$ are the independence and strong independence numbers of $G_t$. Simultaneously, the pseudo-regret against stochastic losses satisfies:*

$$R_T \leq \inf_{0 \leq n \leq T} \left\{ \max_{S \subset V : |S| = \tilde{A}_n} \left\{ \sum_{i \in S : \Delta_i > 0} \frac{4\beta \ln^2 T}{\Delta_i} \right\} + n \right\}$$
$$+ 2\ln T + \sum_{i : \Delta_i > 0} \frac{16K}{\Delta_i} + \frac{161\beta K^{3/2}}{\Delta_{min}^2} \ln\left( \frac{\sqrt{\beta}K}{\Delta_{min}} \right).,$$

*where $\tilde{A}_n$ is the $n^{th}$ largest element in the set containing the strong independence number of all the $G_t$, for $t \leq T$.*

A proof of this theorem is provided in Appendix E.

In the adversarial regime, adapting to graphs that change over time is seamless and does not come at any cost, as using a sequence of fixed graphs exactly recovers the bound of Theorem 2. In the stochastic regime, using $\lambda = 1$ allows us to obtain the same tight constants as in Corollary 1, and only comes at the cost of a multiplicative $\sqrt{K}$ factor in the last term of the bound. Furthermore, the first term of the bound is a sum over the $\tilde{A}_n$ arms that have the smallest non-zero suboptimality gaps. In the case of undirected graphs, if we upper bound the infimum by taking $n = 0$, we have $\tilde{\alpha}_0 = \max_{t>1}\{\alpha(G_t)\}$, which matches the dependency on gaps achieved by Cohen et al. (2016), who got an $O\left(\max_{S \subset V \setminus \{i^*\} : |S| = O(\alpha)} \sum_{i \in S} \frac{\ln T}{\Delta_i}\right)$ bound. This trick is particularly useful if most of the graphs have a small strong independence number and very few have a large independence number, as we can consider the graphs that have a large independence number separately at the cost of an additive constant and in return the dominating term will scale with the strong independence number of the remaining graphs, which may be much smaller.

# 7 Conclusion

Erez and Koren (2021) left open the following questions: is it possible to achieve best-of-both-worlds regret bounds in terms of the independence number, and can the dependence on $T$ in their regret bounds be improved? We partially answered these questions with the EXP3.G++ algorithm and derived near-optimal best-of-both-worlds guarantees for directed feedback graphs. Our regret bounds depend on the independence number of the feedback graphs and improve upon the results of Erez and Koren (2021) by poly-logarithmic factors in both the adversarial and stochastic regimes. Furthermore, we extended our results to time-varying feedback graphs with a computationally efficient algorithm.

## Acknowledgments and Disclosure of Funding

CR and YS acknowledge partial support by the Independent Research Fund Denmark, grant number 9040-00361B. DvdH and NCB gratefully acknowledge partial support from the MIUR PRIN grant Algorithms, Games, and Digital Markets (ALGADIMAR), the EU Horizon 2020 ICT-48 research and innovation action under grant agreement 951847, project ELISE (European Learning and Intelligent Systems Excellence), and the project "One Health Action Hub: University Task Force for the resilience of territorial ecosystems" funded by Università degli Studi di Milano.

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
