## A    Tools to Bound Series

We use the following lemmas to bound series.

**Lemma 2** (Lemma 11 (Seldin and Lugosi, 2017)). *For $\gamma \geq 2$ and $m \geq 1$:*

$$\sum_{k=m}^{n} \frac{1}{k^\gamma} \leq \frac{1}{2m^{\gamma-1}}.$$

**Lemma 3** (Lemma 8 (Seldin et al., 2014)). *For any sequence of non-negative numbers $a_1, a_2, \ldots$, such that $a_1 > 0$ and any power $\gamma \in (0, 1)$ we have:*

$$\sum_{t=1}^{T} \frac{a_t}{\left(\sum_{s=1}^{t} a_s\right)^\gamma} \leq \frac{1}{1-\gamma} \left(\sum_{t=1}^{T} a_t\right)^{1-\gamma}.$$

We also require a variation of this bound to handle the case where the denominator of the sum only sums up to index $t - 1$. The proof of this Lemma follows from Gaillard et al. (2014, Lemma 14) that we generalized to adapt to sequences of $a_t$ that are not restricted to the $[0, 1]$ interval.

**Lemma 4.** *For any sequence $a_1, a_2, \ldots$, such that $a_s \in [1, K]$ for all $s$, we have:*

$$\sum_{t=1}^{T} \frac{a_t}{\sqrt{K + \sum_{s<t} a_s}} \leq 2\sqrt{\sum_{t=1}^{T} a_t} + \sqrt{K}.$$

*Proof.* Let $s_t = \sum_{n=1}^{t} a_t$, and define $s_0 := 0$. We want to bound $\sum_{t=1}^{T} \frac{a_t}{\sqrt{K + \sum_{s<t} a_s}} = \sum_{t=1}^{T} \frac{a_t}{\sqrt{K + s_{t-1}}}$, where $\frac{1}{\sqrt{K+s}}$ is a decreasing function of $s$. Thus we have:

$$\sum_{t=1}^{T} \frac{a_t}{\sqrt{K + s_{t-1}}} = \sum_{t=1}^{T} \frac{a_t}{\sqrt{K + s_t}} + \sum_{t=1}^{T} a_t \left(\frac{1}{\sqrt{K + s_{t-1}}} - \frac{1}{\sqrt{K + s_t}}\right)$$

$$\leq \sum_{t=1}^{T} \frac{a_t}{\sqrt{s_t}} + K \sum_{t=1}^{T} \left(\frac{1}{\sqrt{K + s_{t-1}}} - \frac{1}{\sqrt{K + s_t}}\right)$$

$$\leq \sum_{t=1}^{T} \frac{a_t}{\sqrt{s_t}} + K \frac{1}{\sqrt{K + s_0}}$$

$$\leq 2\sqrt{s_t} + \sqrt{K},$$

where we use Lemma 3 in the last step. $\qquad\square$

**Lemma 5** (Lemma 3 (Thune and Seldin, 2018)). *For $c > 0$ we have*

$$\sum_{t=1}^{\infty} e^{-c\sqrt{t}} \leq \frac{2}{c^2} \qquad and \qquad \sum_{t=1}^{\infty} e^{-ct} \leq \frac{1}{c}.$$

## B    Analysis of the Adversarial Regime

We follow the proof structure of Theorem 2 from Alon et al. (2015), and use Lemma 7 from Seldin and Slivkins (2014) where $X_{t,i} = \tilde{\ell}_{t,i}$ for all $t, i$ as a base for the analysis of EXP3.

**Lemma 6** (Lemma 7 (Seldin and Slivkins, 2014)). *For any $K$ sequences of non-negative numbers $X_{1,i}, X_{2,i}, \ldots$ indexed by $i \in [K]$, and any non-increasing positive sequence $\eta_1, \eta_2, \ldots$, for $q_{t,i} = \frac{\exp(-\eta_t \sum_{s=1}^{t-1} X_{s,i})}{\sum_{j \in [K]} \exp(-\eta_t \sum_{s=1}^{t-1} X_{s,j})}$ (assuming for $t = 1$ the sum in the exponent is 0) we have:*

$$\sum_{t=1}^{T} \sum_{i=1}^{K} q_{t,i} X_{t,i} - \min_{k \in [K]} \sum_{t=1}^{T} X_{t,k} \leq \frac{\ln K}{\eta_T} + \sum_{t=1}^{T} \frac{\eta_t}{2} \left(\sum_{i \in [K]} q_{t,i} X_{t,i}^2\right).$$

We then consider two ways to take advantage of the graph structure. In the first case, we rely on Lemma 5 from Alon et al. (2015) in order to derive a bound that scales with the independence number.

**Lemma 7** (Lemma 5 (Alon et al., 2015)). *Let $G = (V, E)$ be a directed graph with $|V| = K$, in which each node $i \in V$ is assigned a positive weight $w_i$. Assume that $\sum_{i \in V} w_i \leq 1$, and that $w_i \geq \epsilon$ for all $i \in V$ for some constant $0 < \epsilon < \frac{1}{2}$. Then*

$$\sum_{i \in V} \frac{w_i}{w_i + \sum_{j \in N^{\mathrm{in}}(i)} w_j} \leq 4\alpha \ln \left( \frac{4K}{\alpha \epsilon} \right),$$

*where $\alpha = \alpha(G)$ is the independence number of $G$.*

In the second case, we want to derive a bound that scales with the strong independence number of the graph. To do so, we rely on Lemma 10 from Alon et al. (2017). That Lemma depends on a different graph dependent quantity: the maximum acyclic subgraph of a feedback graph $G$, which is defined by Alon et al. as follows. We show that we can upper bound the maximum acyclic subgraph of any graph $G$ in terms of its strong independence number.

**Definition 2.** *Given a directed graph $G = (V, E)$, an acyclic subgraph of $G$ is any $G' = (V', E')$ such that $V' \subseteq V$ and $E' = E \cap (V' \times V')$, with no (directed) cycles. We denote by $\mathrm{mas}(G) = |V'|$ the maximum size of such a $V'$.*

A key property of the maximum acyclic subgraph is that for any graph $G$, $\alpha(G) \leq \mathrm{mas}(G)$ and for undirected graphs, $\alpha(G) = \mathrm{mas}(G)$ (Alon et al., 2017). We now show that for any directed graph $G$, the maximum acyclic subset of $G$ can be upper bounded by its strong independence number.

**Proposition 4.** *Let $G = (V, E)$ be a directed graph. $\mathrm{mas}(G) \leq \tilde{\alpha}(G)$.*

*Proof.* Let $G' = (V', E')$ be an acyclic subgraph of $G$, where $V' \subseteq V$ and $E' = E \cap (V' \times V')$. For any $i, j \in V'$, we know that $(i, j) \notin E'$ or $(j, i) \notin E'$, otherwise $i$ and $j$ would be part of a cycle which contradicts the definition of $G'$. Thus $i$ and $j$ are strongly independent and $V'$ is a strongly independent set. As this holds for all acyclic subgraphs of $G$, we deduce that $\mathrm{mas}(G) \leq \tilde{\alpha}(G)$ which finishes the proof. $\square$

This characterization allows us to use the following lemma and derive bounds that scale with the strong independence number.

**Lemma 8** (Lemma 10 Alon et al. (2017)). *let $G = (V, E)$ be a directed graph with vertex set $V = \{1, \ldots, K\}$, and arc set $V$. Then, for any distribution $p$ over $V$ we have:*

$$\sum_{i=1}^{K} \frac{p_i}{p_i + \sum_{j \in N^{\mathrm{in}}(i)} p_j} \leq \mathrm{mas}(G).$$

With those results, we can move on to the proof of Theorem 2.

*Proof of Theorem 2.* Without loss of generality, we assume that $K \geq 2$.

Recall that the algorithm initializes by playing each arm once, which adds at most $K$ to the regret. The EXP3 part of the analysis starts from round $K + 1$. We can upper trivially upper bound the first $K$ rounds by 1 and then analyse the algorithm from round $t = K + 1$. Precisely, we bound the pseudo-regret as:

$$\mathcal{R}_T = \mathbb{E}\left[\sum_{t=1}^{T} \ell_{t, I_t}\right] - \min_i \mathbb{E}\left[\sum_{t=1}^{T} \ell_{t, i}\right]$$

$$\leq K + \mathbb{E}\left[\sum_{t=K+1}^{T} \ell_{t, I_t}\right] - \min_i \mathbb{E}\left[\sum_{t=K+1}^{T} \ell_{t, i}\right]$$

$$= K + \mathbb{E}\left[\sum_{t=K+1}^{T} \sum_{i=1}^{K} p_{t,i} \mathbb{E}_t\left[\tilde{\ell}_{t,i}\right] - \sum_{t=K+1}^{T} \mathbb{E}_t\left[\tilde{\ell}_{t,i^*}\right]\right]$$

$$\leq K + \mathbb{E}\left[\sum_{t=K+1}^{T} \sum_{i=1}^{K} q_{t,i} \mathbb{E}_t\left[\tilde{\ell}_{t,i}\right] - \sum_{t=K+1}^{T} \mathbb{E}_t\left[\tilde{\ell}_{t,i^*}\right]\right] + \mathbb{E}\left[\sum_{t=K+1}^{T} \sum_{i=1}^{K} \varepsilon_{t,i} \mathbb{E}_t\left[\tilde{\ell}_{t,i}\right]\right], \quad (9)$$

where $i^* = \arg\min\left\{\sum_{t=K+1}^{T}\mathbb{E}_t\left[\tilde{\ell}_{t,i^*}\right]\right\}$, and $\mathbb{E}_t\left[\tilde{\ell}_{t,i}\right] = \ell_{t,i}$. Equation (9) follows from $p_{t,i} \leq q_{t,i} + \varepsilon_{t,i}$. We can consider the contribution of $q_{t,i}$ and $\varepsilon_{t,i}$ separately.

We recall that the learning rate is defined from index $t \geq K+1$ by:

$$\eta_t = \sqrt{\frac{\ln K}{2\sum_{s=K}^{t-1}\theta_s}}, \quad \text{where } \theta_t = \sum_{i \in V}\frac{p_{t,i}}{P_{t,i}}.$$

As the quantities $p_{t,i}$ are not defined for $t \geq K$, we set $\theta_K := K$ to ensure that the learning rate is well defined and non-increasing at all the rounds where we use exponential weights. As the learning rate is a random variable, we have:

$$\mathcal{R}_T \leq K + \mathbb{E}\left[\sum_{t=K+1}^{T}\sum_{i=1}^{K}q_{t,i}\mathbb{E}_t\left[\tilde{\ell}_{t,i}\right] - \sum_{t=K+1}^{T}\mathbb{E}_t\left[\tilde{\ell}_{t,i^*}\right]\right] + \mathbb{E}\left[\sum_{t=K+1}^{T}\sum_{i=1}^{K}\varepsilon_{t,i}\mathbb{E}_t\left[\tilde{\ell}_{t,i}\right]\right]$$

$$\leq K + \mathbb{E}\left[\mathbb{E}_t\left[\frac{\ln K}{\eta_T}\right]\right] + \mathbb{E}\left[\sum_{t=K+1}^{T}\mathbb{E}_t\left[\sum_{i \in V}\frac{\eta_t}{2}\frac{q_{t,i}}{P_{t,i}}\right]\right] + \mathbb{E}\left[\sum_{t=K+1}^{T}\sum_{i=1}^{K}\mathbb{E}_t\left[\varepsilon_{t,i}\tilde{\ell}_{t,i}\right]\right]. \quad (10)$$

We now want to bound each term as a function of the $\theta_t$.

The first term becomes:

$$\mathbb{E}\left[\mathbb{E}_t\left[\frac{\ln K}{\eta_T}\right]\right] \leq \sqrt{2\ln K}\,\mathbb{E}\left[\mathbb{E}_t\left[\sqrt{\sum_{s=K}^{T-1}\theta_s}\right]\right].$$

To bound the second term, we first note that using $\frac{1}{2K}$ as an upper bound on $\varepsilon_t$, we ensure that for all $t$ and $i$, $p_{t,i} \geq (1 - \sum_{j \in V}\varepsilon_{t,j})q_{t,i} \geq \frac{1}{2}q_{t,i}$ which gives:

$$\sum_{i \in V}\frac{q_{t,i}}{P_{t,i}} \leq 2\sum_{i \in V}\frac{p_{t,i}}{P_{t,i}} = 2\theta_t.$$

Then, the second term can be bounded as:

$$\mathbb{E}\left[\sum_{t=K+1}^{T}\mathbb{E}_t\left[\sum_{i \in V}\frac{\eta_t}{2}\frac{q_{t,i}}{P_{t,i}}\right]\right] \leq \mathbb{E}\left[\sum_{t=K+1}^{T}\mathbb{E}_t\left[\eta_t\sum_{i \in V}\frac{p_{t,i}}{P_{t,i}}\right]\right]$$

$$\leq \sqrt{\ln K}\,\mathbb{E}\left[\sum_{t=K+1}^{T}\mathbb{E}_t\left[\frac{\theta_t}{\sqrt{2\sum_{s=K}^{t-1}\theta_s}}\right]\right]$$

$$\leq \sqrt{2\ln K}\,\mathbb{E}\left[\mathbb{E}_t\left[\sqrt{\sum_{t=K+1}^{T}\theta_t}\right]\right] + \sqrt{K}, \quad (11)$$

where equation (11) follows from Lemma 4.

For the last term, we recall that we bounded $\varepsilon_{t,i} \leq \frac{1}{2}\sqrt{\frac{\lambda\ln K}{tK^2}}$, and we have:

$$\mathbb{E}\left[\sum_{t=K+1}^{T}\sum_{i=1}^{K}\varepsilon_{t,i}\mathbb{E}_t\left[\tilde{\ell}_{t,i}\right]\right] \leq \sum_{t=K+1}^{T}\mathbb{E}\left[\sum_{i=1}^{K}\frac{1}{2}\sqrt{\frac{\lambda\ln K}{tK^2}}\mathbb{E}_t\left[\tilde{\ell}_{t,i}\right]\right]$$

$$\leq \sum_{t=1}^{T}\frac{1}{2}\sqrt{\frac{\lambda\ln K}{t}}$$

$$\leq \sqrt{\lambda T\ln K}.$$

Using those three bounds in equation 10 gives:

$$\mathcal{R}_T \leq 2K + 2\sqrt{2\ln K}\, \mathbb{E}\left[\mathbb{E}_t\left[\sqrt{\sum_{t=K+1}^{T} \theta_t}\right]\right] + \sqrt{\lambda T \ln K}. \tag{12}$$

All that remains is to bound $\theta$ and $\lambda$. To obtain the first part of the bound, we use Proposition 4 and Lemma 8, and deduce that for all $t \leq K + 1$ :

$$\theta_t \leq \tilde{\alpha} \text{ , which gives } \sqrt{\sum_{t=K+1}^{T} \theta_t} \leq \sqrt{\tilde{\alpha} T}. \tag{13}$$

Using $\lambda \leq \tilde{\alpha}$, we deduce that:

$$\mathcal{R}_T \leq 2K + 2\sqrt{2\ln K}\,\sqrt{\tilde{\alpha}T} + \sqrt{\tilde{\alpha}T\ln K}$$
$$\leq 4\sqrt{\tilde{\alpha}T\ln K} + 2K. \tag{14}$$

For the second part of the bound, we use Lemma 7 and recall that for all $t$ and $i$, $\varepsilon_{t,i} \geq \frac{4}{t^2}$. We deduce the following upper bound:

$$\theta_t \leq 4\alpha\ln\left(\frac{t^2 K}{\alpha}\right) \leq 8\alpha\ln(KT) \text{ , which gives } \sqrt{\sum_{t=K+1}^{T} \theta_t} \leq \sqrt{8\alpha T\ln(KT)}.$$

Using $\lambda \leq \alpha\ln T$, we deduce that:

$$\mathcal{R}_T \leq 2K + 2\sqrt{2\ln K}\,\sqrt{8\alpha T\ln(KT)} + \sqrt{\ln K}\sqrt{\ln T}\sqrt{\alpha T}$$
$$\leq 9\sqrt{\ln K}\,\sqrt{\ln(KT)}\sqrt{\alpha T} + 2K, \tag{15}$$

Taking the minimum between equations (14) and (15) finishes the proof.

$$\square$$

## C  Properties of the Gaps Estimates

In this section, we provide upper and lower high probability bounds for the estimates of the suboptimality gaps. We decompose the proof of Lemma 1 in two parts.

### C.1  Upper bound

We start by deriving a high probability upper bound. For this bound, we have to be careful with the fact that the gap estimates are clipped in the $[0, 1]$ interval. We first upper derive bounds on UCB and LCB.

**Lemma 9.** *The confidence intervals satisfy:*

$$\mathbb{P}\left[UCB_{t,i} \leq \mu_i\right] \leq \frac{1}{KT^{\gamma-1}}$$

$$\text{and} \quad \mathbb{P}\left[LCB_{t,i} \geq \mu_i\right] \leq \frac{1}{KT^{\gamma-1}}.$$

*Proof.* Let $\overline{UCB}_t$ and $\overline{LCB}_t$ be the non clipped versions of the $UCB_t$ and $LCB_t$. In other words, for all $i$ and $t$:

$$\overline{UCB}_{t,i} = \frac{\hat{L}_{t-1,i}}{O_{t-1,i}} + \sqrt{\frac{\gamma\ln\left(tK^{1/\gamma}\right)}{2O_{t-1,i}}}$$

$$\text{and} \quad \overline{LCB}_{t,i} = \frac{\hat{L}_{t-1,i}}{O_{t-1,i}} - \sqrt{\frac{\gamma\ln\left(tK^{1/\gamma}\right)}{2O_{t-1,i}}}.$$

Then, through standard UCB analysis using Hoeffding's inequality (see for example Seldin and Lugosi (2017)), we have:

$$\mathbb{P}\left[\overline{\mathrm{UCB}}_{t,i} \leq \mu_i\right] \leq \frac{1}{KT^{\gamma-1}}$$

$$\text{and} \qquad \mathbb{P}\left[\overline{\mathrm{LCB}}_{t,i} \geq \mu_i\right] \leq \frac{1}{KT^{\gamma-1}}.$$

By definition, we have $\mathrm{UCB}_{t,i} = \min\left\{1, \overline{\mathrm{UCB}}_{t,i}\right\} \leq \overline{\mathrm{UCB}}_{t,i}$ and $\mathrm{LCB}_{t,i} = \max\left\{0, \overline{\mathrm{LCB}}_{t,i}\right\} \geq \overline{\mathrm{LCB}}_{t,i}$, so:

$$\mathbb{P}\left[\mathrm{UCB}_{t,i} \leq \mu_i\right] \leq \mathbb{P}\left[\overline{\mathrm{UCB}}_{t,i} \leq \mu_i\right] \leq \frac{1}{KT^{\gamma-1}}$$

$$\text{and} \qquad \mathbb{P}\left[\mathrm{LCB}_{t,i} \geq \mu_i\right] \leq \mathbb{P}\left[\overline{\mathrm{LCB}}_{t,i} \geq \mu_i\right] \leq \frac{1}{KT^{\gamma-1}}.$$

$\square$

Using this result, we can move on to bound the gap estimates.

*Proof of the first part of Lemma 1.* We recall that $\hat{\Delta}_{t,i} = \max\left\{0, \mathrm{LCB}_{t,i} - \min_{j \neq i} \mathrm{UCB}_{t,j}\right\}$. Then using Lemma 9, we have:

$$\begin{aligned}
\mathbb{P}\left[\hat{\Delta}_{t,i} \geq \overline{\Delta_i}\right] &= \mathbb{P}\left[\mathrm{LCB}_{t,i} - \min_{j \neq i} \mathrm{UCB}_{t,j} \geq \overline{\Delta_i}\right] \\
&\leq \mathbb{P}\left[\mathrm{LCB}_{t,i} - \min_{j \neq i} \mathrm{UCB}_{t,j} \geq \Delta_i\right] \\
&\leq \mathbb{P}\left[\mathrm{LCB}_{t,i} \geq \mu_i\right] + \sum_{j \neq i} \mathbb{P}\left[\mathrm{UCB}_{t,j} \leq \mu_j\right] \\
&\leq K \frac{1}{Kt^{\gamma-1}} = \frac{1}{t^{\gamma-1}},
\end{aligned}$$

where the first step takes advantage of the fact that $\overline{\Delta}_i > 0$ for all $i$, allowing to remove the maximum. The second step relies on $\Delta_i \leq \overline{\Delta}_i$, and we finish the proof with a union bound and applying Lemma 9. $\square$

## C.2 Lower bound

To derive a lower bound on the gap estimates and prove the second part of Lemma 1, we start by proving some intermediate results. recall that we use $o_{t,i}$ to lower bound the probability of observing the loss of arm $i$ at round $t$, and that by construction we have for all $t, i$:

$$o_{t,i} = \min\left\{\frac{1}{2K}, \frac{1}{2}\sqrt{\frac{\lambda \ln K}{tK^2}}, \frac{\beta \ln t}{t\hat{\Delta}_{t,i}^2}\right\}.$$

We also recall that for all $i$ such that $\Delta_i > 0$, we defined $t_{\min}(i)$ as:

$$t_{\min}(i) = \max\left\{t \geq 0 : \frac{1}{2}\sqrt{\frac{\lambda \ln K}{tK^2}} \leq \frac{\beta \ln t}{t\Delta_i^2}\right\}.$$

**A lower bound for $o_{t,i}$.** As $\hat{\Delta}_{t,i}$ is a random variable, we derive a high probability lower bounds on $o_{t,i}$.

**Definition 3.** *We define the following events:*

$$\mathcal{E}(i,t) = \left\{\forall s \in [K+1, t] : o_{s,i} \geq \frac{\beta \ln t}{t\Delta_i^2}\right\},$$

$$\mathcal{E}(i^*, i, t) = \left\{\forall s \in [K+1, t] : o_{s,i^*} \geq \frac{\beta \ln t}{t\Delta_i^2}\right\},$$

*where $i^*$ is an optimal arm and $i$ a suboptimal arm.*

Note that the second event lower bounds the rate at which observations on optimal arm $i^*$ are gathered in terms of the gap with the suboptimal arm $i$.

**Lemma 10.** *For any $i$ suboptimal arm and $i^*$ optimal arm, and $t \geq t_{\min}(i)$ and $\gamma \geq 3$, we have:*

$$\mathbb{P}\left[\overline{\mathcal{E}(i,t)}\right] \leq \left(\frac{\ln t}{t\Delta_i^2}\right)^{\gamma-2},$$

$$\mathbb{P}\left[\overline{\mathcal{E}(i^*,i,t)}\right] \leq \left(\frac{\ln t}{t\Delta_i^2}\right)^{\gamma-2}.$$

*Proof of Lemma 10.* The proof is very similar for the two inequalities. By definition for all $s$ and $i$, we have $\hat{\Delta}_{s,i} \leq 1$. Thus, $\frac{\beta \ln s}{s\hat{\Delta}_{s,i}^2} \geq \frac{\beta \ln s}{s}$. Then for $s \in \left[K+1, \frac{t\Delta_i^2}{\ln t}\right]$, we have $\frac{\beta \ln s}{s} \geq \frac{\beta \ln s \ln t}{t\Delta_i^2} \geq \frac{\beta \ln t}{t\Delta_i^2}$, as $s > K \geq 2$, so $\ln s \geq 1$. Furthermore, as $t \geq t_{\min}(i)$ then for all $s \in [K+1, t]$, we have $\frac{1}{2}\sqrt{\frac{\lambda \ln K}{sK^2}} \geq \frac{1}{2}\sqrt{\frac{\lambda \ln K}{tK^2}} \geq \frac{\beta \ln t}{t\Delta_i^2}$ and $\frac{1}{2K} \geq \frac{\beta \ln t}{t\Delta_i^2}$. We deduce:

$$\mathbb{P}\left[\overline{\mathcal{E}(i,t)}\right] = \mathbb{P}\left[\exists s \in \left[\frac{t\Delta_i^2}{\ln t}, t\right] : o_{s,i} \leq \frac{\beta \ln t}{t\Delta_i^2}\right]$$

$$\leq \mathbb{P}\left[\exists s \in \left[\frac{t\Delta_i^2}{\ln t}, t\right] : \hat{\Delta}_{s,i} \geq \Delta_i\right] \qquad (16)$$

$$\leq \sum_{s=\frac{t\Delta_i^2}{\ln t}} \frac{1}{s^{\gamma-1}}$$

$$\leq \frac{1}{2}\left(\frac{\ln t}{t\Delta_i^2}\right)^{\gamma-2},$$

where the last summation follows from Lemma 2. The proof of the second inequality is similar, Equation (16) only requiring the extra step:

$$\mathbb{P}\left[\exists s \in \left[\frac{t\Delta_{i^*}^2}{\ln t}, t\right] : \hat{\Delta}_{s,i} \geq \Delta_i\right] \leq \mathbb{P}\left[\exists s \in \left[\frac{t\Delta_{i^*}^2}{\ln t}, t\right] : \hat{\Delta}_{s,i} \geq \overline{\Delta_{i^*}}\right],$$

which follows from $\Delta_i \geq \Delta_{\min} = \overline{\Delta_{i^*}}$ as $i$ is a suboptimal arm and $i^*$ is an optimal arm. $\qquad \square$

**A lower bound for $O_{t,i}$** We now want to lower bound the number of observations of an arm up to round $t$. We rely on the following concentration inequality.

**Theorem 5** (Theorem 8 (Seldin and Lugosi, 2017)). *Let $X_1, \ldots, X_n$ be Bernoulli random variables adapted to filtration $\mathcal{F}_1, \ldots, \mathcal{F}_n$ (in particular, $X_s$ may depend on $X_1, \ldots, X_{s-1}$). Let $\mathcal{E}_\lambda$ be the event $\mathcal{E}_\lambda = \{\forall s : \mathbb{E}[X_s|\mathcal{F}_{s-1}] \geq \lambda\}$. Then,*

$$\mathbb{P}\left[\left(\sum_{s=1}^n X_s \leq \frac{1}{2}n\lambda\right) \wedge \mathcal{E}_\lambda\right] \leq e^{-n\lambda/8}.$$

We recall that the first $K$ rounds of the algorithm are deterministic, and that each arm is observed at least once. We use $O_{[K+1:t],i}$ to refer to the number of observations from rounds $K+1$ to $t$, and we note that $O_{t,i} \geq O_{[K+1:t],i} + 1$. We have:

$$\mathbb{P}\left[O_{t,i} \leq \frac{\beta \ln t}{2\Delta_i^2}\right] \leq \mathbb{P}\left[O_{[K+1:t],i} \leq \frac{\beta \ln t}{2\Delta_i^2} - 1\right]$$

$$\leq \mathbb{P}\left[O_{[K+1:t],i} \leq \frac{\beta \ln t}{2\Delta_i^2}\frac{t-K}{t}\right],$$

where the second step follows from, $\frac{\beta \ln t}{2\Delta_i^2} - 1 \leq \frac{\beta \ln t}{2\Delta_i^2}\frac{t-K}{t} \Leftrightarrow \frac{K\beta \ln t}{2t\Delta_i^2} \leq 1$, which is true for $t \geq t_{\min}(i)$ as

$$\frac{K\beta \ln t}{2t\Delta_i^2} \leq \frac{K\beta \ln t}{2\Delta_i^2}\frac{\Delta_i^4\lambda \ln K}{4K^2\beta^2 \ln^2 t} \leq \frac{\Delta_i^2 \ln K}{8\ln t} \leq \frac{\Delta_i^2}{8} \leq 1.$$

We can apply Theorem 5 on the $t - K$ random variables $\mathbb{1}\left[i \in N^{\text{out}}(I_s)\right]$ for $s \in [K+1, t]$ and we get:

$$\mathbb{P}\left[O_{t,i} \leq \frac{\beta \ln t}{2\Delta_i^2}\right] \leq \mathbb{P}\left[\left(O_{[K+1:t],i} \leq \frac{\beta \ln t}{2\Delta_i^2}\frac{t-K}{t}\right) \wedge \mathcal{E}_{t,i}\right] + \mathbb{P}\left[\overline{\mathcal{E}_{t,i}}\right]$$

$$\leq e^{-\frac{t-K}{t}\frac{\beta \ln t}{8\Delta_i^2}} + \frac{1}{2}\left(\frac{\ln t}{t\Delta_i^2}\right)^{\gamma-2}$$

$$\leq e^{-\frac{3}{4}\frac{\beta \ln t}{8\Delta_i^2}} + \frac{1}{2}\left(\frac{\ln t}{t\Delta_i^2}\right)^{\gamma-2}$$

$$\leq \left(\frac{1}{t}\right)^{\beta/10} + \frac{1}{2}\left(\frac{\ln t}{t\Delta_i^2}\right)^{\gamma-2},$$

where we use that $t \geq t_{\min}(i) \geq 4K$, so $\frac{t-K}{t} \geq \frac{3}{4}$.

**A lower bound for $\hat{\Delta}_{t,i}$**   Using Lemma 9, we know that the upper and lower confidence bounds satisfy: $\mathbb{P}\left[\text{UCB}_{t,i^*} \leq \mu_{i^*} \vee \text{LCB}_{t,i} \geq \mu_i\right] \leq \frac{2}{Kt^{\gamma-1}}$. Then assuming that $\text{UCB}_{t,i^*} \geq \mu_{i^*}$ and $\text{LCB}_{t,i} \leq \mu_i$, we have:

$$\hat{\Delta}_{t,i} \geq \text{LCB}_{t,i} - \min_{j \neq i} \text{UCB}_{t,i}$$

$$\geq \text{LCB}_{t,i} - \text{UCB}_{t,i^*}$$

$$\geq \frac{\hat{L}_{t-1,i}}{O_{t-1,i}} - \sqrt{\frac{\gamma \ln\left(tK^{1/\gamma}\right)}{2O_{t-1,i}}} - \frac{\hat{L}_{t-1,i^*}}{O_{t-1,i^*}} - \sqrt{\frac{\gamma \ln\left(tK^{1/\gamma}\right)}{2O_{t-1,i^*}}}$$

$$= \frac{\hat{L}_{t-1,i}}{O_{t-1,i}} + \sqrt{\frac{\gamma \ln\left(tK^{1/\gamma}\right)}{2O_{t-1,i}}} - 2\sqrt{\frac{\gamma \ln\left(tK^{1/\gamma}\right)}{2O_{t-1,i}}}$$

$$- \left(\frac{\hat{L}_{t-1,i^*}}{O_{t-1,i^*}} - \sqrt{\frac{\gamma \ln\left(tK^{1/\gamma}\right)}{2O_{t-1,i^*}}}\right) - 2\sqrt{\frac{\gamma \ln\left(tK^{1/\gamma}\right)}{2O_{t-1,i^*}}}$$

$$= \text{UCB}_{t,i} - \text{LCB}_{t,i^*} - 2\sqrt{\frac{\gamma \ln\left(tK^{1/\gamma}\right)}{2O_{t-1,i}}} - 2\sqrt{\frac{\gamma \ln\left(tK^{1/\gamma}\right)}{2O_{t-1,i^*}}}$$

$$\geq \Delta_i - 2\sqrt{\frac{\gamma \ln\left(tK^{1/\gamma}\right)}{2O_{t-1,i}}} - 2\sqrt{\frac{\gamma \ln\left(tK^{1/\gamma}\right)}{2O_{t-1,i^*}}}.$$

Using the previously derived high probability bounds, assuming that $O_{t,i} \geq \frac{\beta \ln t}{2\Delta_i^2}$ and $O_{t,i^*} \geq \frac{\beta \ln t}{2\Delta_i^2}$, and using that $t \geq t_{\min}(i) \geq K$, we have:

$$\hat{\Delta}_{t,i} \geq \Delta_i - 2\sqrt{\frac{\gamma \ln\left(tK^{1/\gamma}\right)}{2O_{t-1,i}}} - 2\sqrt{\frac{\gamma \ln\left(tK^{1/\gamma}\right)}{2O_{t-1,i^*}}}$$

$$\geq \Delta_i - 4\sqrt{\frac{2\Delta_i^2 \gamma \ln\left(tK^{1/\gamma}\right)}{2\beta \ln t}}$$

$$\geq \Delta_i - 4\sqrt{\frac{\Delta_i^2(\gamma+1)\ln\left(tK^{1/\gamma}\right)}{\beta \ln t}} \qquad (17)$$

$$= \Delta_i\left(1 - 4\sqrt{\frac{\gamma+1}{\beta}}\right),$$

where equation (17) follows from $t \geq K$, so $\gamma \ln\left(tK^{1/\gamma}\right) = \gamma \ln\left(t^\gamma K\right) \leq \ln\left(t^\gamma t\right) = (\gamma+1)\ln t$. Using that $\beta \geq 64(\gamma+1)$, we have:

$$\mathbb{P}\left[\hat{\Delta}_{t,i} \leq \frac{1}{2}\Delta_i\right] \leq \left(\frac{\ln t}{t\Delta_i^2}\right)^{\gamma-2} + \frac{2}{Kt^{\gamma-1}} + 2\left(\frac{1}{t}\right)^{\beta/10}.$$

# D  Analysis of the Stochastic Regime

In the stochastic regime, we decompose the regret bound into three terms that we bound separately. First, during the initial $t_{\min} = \max_{i:\Delta_i>0}\{t_{\min}(i)\}$ rounds we use the adversarial bound. Then, in the remaining rounds we bound the contribution of the exponential weights and of the exploration separately.

## D.1  Control over the Initial Rounds

We start by deriving a time independent upper bound on $t_{\min}(i)$ for all vertices $i$ such that $\Delta_i > 0$.

**Proposition 5.** *For any constant $c > e^2$, we have:*

$$\max_t \left\{t \le c(\ln t)^2\right\} \le 25c\,(\ln c)^2.$$

*Proof.* First, we note that for $t = 3$,

$$c(\ln t)^2 \ge e^2(\ln 3)^2 \ge 3 = t,$$

so the inequality is fulfilled at $t = 3$.

Furthermore, $(c(\ln t)^2)' = 2c\frac{\ln t}{t}$ is a decreasing function of $t$ for $t \ge e$ and such that $\lim_{t\to\infty} 2c\frac{\ln t}{t} = 0$, whereas $(t)' = 1$ is constant. Thus, $\max_t\left\{t \le c(\ln t)^2\right\}$ exists and is solution of

$$t = c\,(\ln t)^2.$$

Let's upper bound this $t$. We denote by $W_{-1}$ the product log function. Then we have:

$$
\begin{aligned}
t &= c\,(\ln t)^2 \\
\sqrt{t} &= \sqrt{c}\;\ln t \\
\sqrt{t} &= 2\sqrt{c}\ln(\sqrt{t}) \\
x &= b\ln(x) && b = 2\sqrt{c},\ x = \sqrt{t} \\
x &= -\,bW_{-1}\left(-\frac{1}{b}\right) && \text{for } b \ge e.
\end{aligned}
$$

By Chatzigeorgiou (2013, Theorem 1), we have for $b \ge e$:

$$-bW_{-1}\left(-\frac{1}{b}\right) = -\,bW_{-1}\left(-\exp\left(-\ln\left(\frac{b}{e}\right)-1\right)\right) \le b\left(1 + \sqrt{2\ln\left(\frac{b}{e}\right)} + \ln\left(\frac{b}{e}\right)\right).$$

Thus, we have that for $c \ge e^2$,

$$
\begin{aligned}
t &\le 4c\left(1 + \sqrt{2\ln\left(\frac{2\sqrt{c}}{e}\right)} + \ln\left(\frac{2\sqrt{c}}{e}\right)\right)^2 \\
&\le 4c\,(1 + 2\ln c)^2 \\
&\le 25c\,(\ln c)^2,
\end{aligned}
$$

where the last step follows from $(1 + 2\ln c)^2 \le (\frac{1}{2}\ln(e^2) + 2(\ln c))^2 \le (2.5\ln c)^2 = 6.25\,(\ln c)^2$. $\qquad\square$

**Proposition 6.** *Under the conditions of Lemma 1 with $\gamma = 4$, $\beta = 320$ and $\lambda \in [1, K]$, the contribution of the initial $t_{\min}$ rounds to the regret can be bounded as:*

$$R_{t_{\min}} \le \min\left\{\frac{160\beta K}{\Delta_{min}^2}\sqrt{\frac{\tilde{\alpha}}{\lambda}}\ln\left(\frac{\sqrt{\beta}K}{\Delta_{min}}\right), \frac{1019\beta K}{\Delta_{min}^2}\sqrt{\frac{\alpha}{\lambda}}\left(\ln\left(\frac{\beta K}{\Delta_{min}}\right)\right)^{3/2}\right\} + 2K.$$

*Proof.* By definition, we have $t_{\min} = \max\left\{t \geq 0 : \frac{1}{2}\sqrt{\frac{\lambda \ln K}{tK^2}} \leq \frac{\beta \ln t}{t\Delta_{\min}^2}\right\}$. By proposition 5, we have that $t_{\min} \leq 25d(\ln d)^2$, where $d = \frac{4\beta^2 K^2}{\lambda \ln K \Delta_{\min}^4}$.

Then, we can use the first half Theorem 2 and deduce that:

$$
\begin{aligned}
R_{t_{\min}} &\leq 4\sqrt{\tilde{\alpha} \ln K \, t_{\min}} + 2K \\
&\leq 4\sqrt{\tilde{\alpha} \ln K \, 25d} \ln(d) + 2K \\
&= 4\sqrt{\tilde{\alpha} \ln K \, 25 \frac{4\beta^2 K^2}{\lambda \ln K \Delta_{\min}^4}} \ln\left(\frac{4\beta^2 K^2}{\lambda \ln K \Delta_{\min}^4}\right) + 2K \\
&\leq \frac{160\beta K}{\Delta_{\min}^2}\sqrt{\frac{\tilde{\alpha}}{\lambda}} \ln\left(\frac{\sqrt{\beta}K}{\Delta_{\min}}\right) + 2K.
\end{aligned}
$$

For the second part of the bound, we use the second part of Theorem 2, and we deduce:

$$
\begin{aligned}
R_{t_{\min}} &\leq 9\sqrt{\alpha t_{\min}}\sqrt{\ln(K t_{\min})}\sqrt{\ln K} + 2K \\
&\leq 9\sqrt{\alpha \ln K \, 25d} \ln(d)\sqrt{\ln(25Kd^2)} + 2K \tag{18} \\
&\leq 9\sqrt{\alpha \ln K \, 25 \frac{4\beta^2 K^2}{\lambda \ln K \Delta_{\min}^4}} \ln\left(\frac{4\beta^2 K^2}{\lambda \ln K \Delta_{\min}^4}\right)\sqrt{\ln\left(25K\left(\frac{4\beta^2 K^2}{\lambda \ln K \Delta_{\min}^4}\right)^2\right)} + 2K \\
&\leq 9 * 5 * 2\frac{\beta K}{\Delta_{\min}^2}\sqrt{\frac{\alpha}{\lambda}} \ln\left(\frac{4\beta^2 K^2}{\lambda \ln K \Delta_{\min}^4}\right)\sqrt{\ln\left(\frac{400\beta^4 K^5}{\Delta_{\min}^8}\right)} + 2K \tag{19} \\
&\leq \frac{1019\beta K}{\Delta_{\min}^2}\sqrt{\frac{\alpha}{\lambda}}\left(\ln\left(\frac{\beta K}{\Delta_{\min}}\right)\right)^{3/2} + 2K.
\end{aligned}
$$

where the equation (18) uses that for $d > 1$ we have $d(\ln d)^2 \leq d^2$, and equation (19) uses that $400 \leq 320^4 \leq \beta^4$.

$\square$

## D.2 Control over the Exponential Weights

Proposition 2 introduced in the proof sketch of Theorem 3 is based on the following result.

**Proposition 7.** *Under the conditions of Lemma 1 with $\gamma = 4$, $\beta = 320$ and $\lambda \in [1, K]$, the sum of exponential weights with sequence of learning rates $\eta_1, \eta_2, \ldots$ of each suboptimal arm $i$ can be bounded as:*

$$
\sum_{t=t_{\min}(i)}^{T} \mathbb{E}\left[q_{t,i}\right] \leq \sum_{t=t_{\min}(i)}^{T}\left(e^{-\frac{1}{2}t\eta_t \Delta_i} + \frac{1}{t}\left(\frac{\lambda \ln K}{4K^2\beta^2} + \frac{1}{K}\right)\right)
$$

To prove this result, we can follow the same derivation as in Seldin and Lugosi (2017). We want to bound the $q_{t,i}$ for all $i$ such that $\Delta_i > 0$ and $t \geq t_{\min}(i)$. First, we note that

$$
\begin{aligned}
q_{t,i} &= \frac{\exp(-\eta_t \tilde{L}_{t,i})}{\sum_{j \in V} \exp(-\eta_t \tilde{L}_{t,j})} \\
&= \frac{\exp(-\eta_t(\tilde{L}_{t,i} - \tilde{L}_{t,i^*}))}{\sum_{j \in V} \exp(-\eta_t(\tilde{L}_{t,j} - \tilde{L}_{t,i^*}))} \\
&\leq \exp(-\eta_t(\tilde{L}_{t,i} - \tilde{L}_{t,i^*})) \\
&:= \exp(-\eta_t \tilde{\Delta}_{t,i}),
\end{aligned}
$$

where $i^*$ is the best arm, and where the inequality holds because one term of the sum is $\exp(-\eta_t(\tilde{L}_{t,i^*} - \tilde{L}_{t,i^*})) = 1$ and the other terms are positive, so the denominator is greater

than 1. We now want to want to ensure that $\tilde{\Delta}_{t,i} := \tilde{L}_{t,i} - \tilde{L}_{t,i^*}$ is close to $t\Delta_i$. To do so, we want to apply a variant of Bernstein's inequality on the martingale sequence difference $t\Delta_i - \tilde{\Delta}_{t,i} = \sum_{s=1}^{t} X_s$, where each single term of the sequence is defined as $X_s = \Delta_i - (\tilde{\ell}_{s,i} - \tilde{\ell}_{s,i^*})$.

**Theorem 6** (Bernstein's inequality for martingales). *Let $X_1, \ldots, X_n$ be a martingale difference sequence with respect to filtration $\mathcal{F}_1, \ldots, \mathcal{F}_n$, where each $X_j$ is bounded from above, and let $S_i = \sum_{j=1}^{i} X_j$ be the associated martingale. Let $\nu_n = \sum_{j=1}^{n} \mathbb{E}\left[(X_j)^2 | \mathcal{F}_{j-1}\right]$ and $\kappa_n = \max_{1 \leq j \leq n} \{X_j\}$. Then, for any $\delta > 0$:*

$$\mathbb{P}\left[\left(S_n \geq \sqrt{2\nu \ln\left(\frac{1}{\delta}\right)} + \frac{\kappa \ln\left(\frac{1}{\delta}\right)}{3}\right) \wedge (\nu_n \leq \nu) \wedge (\kappa_n \leq \kappa)\right] \leq \delta.$$

In order to apply the this theorem, we need to bound $\max_{1 \leq s \leq n} \{X_s\}$ and $\sum_{s=1}^{n} \mathbb{E}\left[(X_s)^2 | \mathcal{F}_{s-1}\right]$.

**Control of** $\max_{1 \leq s \leq t} \{X_s\}$   For each $s$ we have:

$$\begin{aligned}
X_s &= \Delta_i - (\tilde{\ell}_{s,i} - \tilde{\ell}_{s,i^*}) \\
&\leq 1 + \tilde{\ell}_{s,i^*} \\
&\leq 1 + \frac{1}{P_{t,i^*}} \\
&\leq 1 + \max\left\{2K, 2\sqrt{\frac{sK^2}{\lambda \ln K}}, \frac{s\hat{\Delta}_{s,i^*}^2}{\beta \ln s}\right\} \\
&\leq 1.25 \max\left\{2K, 2\sqrt{\frac{sK^2}{\lambda \ln K}}, \frac{s\hat{\Delta}_{s,i^*}^2}{\beta \ln s}\right\}
\end{aligned} \tag{20}$$

where equation (20) holds by definition of $o_{t,i^*}$. Using the same argument as in the proof of Lemma 1, we know that $t \geq t_{\min}(i)$, and if $s \leq \frac{t\Delta_i^2}{\ln t}$ then $\frac{s\hat{\Delta}_i^2}{\ln s} \leq \frac{s}{\beta} \leq \frac{t\Delta_i^2}{\beta \ln t}$ then:

$$\begin{aligned}
\mathbb{P}\left[\exists s \leq t : \max\left\{2K, 2\sqrt{\frac{sK^2}{\lambda \ln K}}, \frac{s\hat{\Delta}_{s,i^*}^2}{\beta \ln s}\right\} \geq \frac{t\Delta_i^2}{\beta \ln t}\right] &= \mathbb{P}\left[\exists s \in \left[\frac{t\Delta_i^2}{\ln t}, t\right] : \Delta_{s,i^*} \geq \Delta_i\right] \\
&\leq \mathbb{P}\left[\exists s \in \left[\frac{t\Delta_i^2}{\ln t}, t\right] : \Delta_{s,i^*} \geq \overline{\Delta_{i^*}}\right],
\end{aligned}$$

because $\Delta_i \geq \Delta_{\min} = \bar{\Delta}_{i^*}$. Let $\kappa_t = \max_{1 \leq s \leq t} \{X_s\}$, and we deduce:

$$\mathbb{P}\left[\kappa_t \geq \frac{1.25 t\Delta_i^2}{\beta \ln t}\right] \leq \mathbb{P}\left[\exists s \in \left[\frac{t\Delta_i^2}{\ln t}, t\right] : \Delta_{s,i^*} \geq \overline{\Delta_{i^*}}\right].$$

**Control of** $\nu_t = \sum_{s=1}^{t} \mathbb{E}\left[(X_s)^2 | \mathcal{F}_{s-1}\right]$   We start by looking at each individual element of the sum.

$$\begin{aligned}
\mathbb{E}\left[(X_s)^2 | \mathcal{F}_{s-1}\right] &= \mathbb{E}\left[(\Delta_i - (\tilde{\ell}_{s,i} - \tilde{\ell}_{s,i^*}))^2 | \mathcal{F}_{s-1}\right] \\
&\leq \mathbb{E}\left[(\tilde{\ell}_{s,i} - \tilde{\ell}_{s,i^*})^2 | \mathcal{F}_{s-1}\right] \\
&\leq \mathbb{E}\left[\tilde{\ell}_{s,i}^2 | \mathcal{F}_{s-1}\right] + \mathbb{E}\left[\tilde{\ell}_{s,i^*}^2 | \mathcal{F}_{s-1}\right],
\end{aligned}$$

where the last equation holds, because for all non-negative $a$ and $b$, we have: $(a - b)^2 \leq a^2 + b^2$. Then, note that:

$$E[\tilde{\ell}_{s,i}^2 | \mathcal{F}_{s-1}] \leq \frac{1}{P_{t,i}},$$

so

$$\mathbb{E}\left[(X_s)^2|\mathcal{F}_{s-1}\right] \le \frac{1}{P_{s,i}} + \frac{1}{P_{s,i^*}}.$$

Using the same argument as before to bound $\frac{1}{P_{s,i^*}}$ and $\frac{1}{P_{s,i}}$, we have:

$$\mathbb{P}\left[\sum_{s=1}^{t}\mathbb{E}\left[(X_s)^2|\mathcal{F}_{s-1}\right] \ge \frac{2t^2\Delta_i^2}{\beta\ln t}\right] \le \mathbb{P}\left[\exists s \in \left[\frac{t\Delta_i^2}{\ln t}, t\right] : \Delta_{s,i^*} \ge \overline{\Delta_{i^*}}\right]$$
$$+ \mathbb{P}\left[\exists s \in \left[\frac{t\Delta_i^2}{\ln t}, t\right] : \Delta_{s,i^*} \ge \overline{\Delta_{i^*}}\right].$$

Noting that the bounds on $\kappa_t$ and $\nu_t$ depend on the same events, we deduce that:

$$\mathbb{P}\left[\left(\kappa_t \ge \frac{1.25t\Delta_i^2}{\beta\ln t}\right) \vee \left(\nu_t \ge \frac{2t^2\Delta_i^2}{\beta\ln t}\right)\right] \le \mathbb{P}\left[\exists s \in \left[\frac{t\Delta_i^2}{\ln t}, t\right] : \Delta_{s,i} \ge \overline{\Delta_i}\right]$$
$$+ \mathbb{P}\left[\exists s \in \left[\frac{t\Delta_i^2}{\ln t}, t\right] : \Delta_{s,i^*} \ge \overline{\Delta_{i^*}}\right]$$
$$\le \frac{1}{t}\frac{\lambda\ln K}{4K^2\beta^2},$$

where for the last step we use that for all $j, k \in V$ and $\gamma = 4$,

$$\mathbb{P}\left[\exists s \in \left[\frac{t\Delta_k^2}{\ln t}, t\right] : \Delta_{s,j} \ge \overline{\Delta_j}\right] \le \sum_{s=\frac{t\Delta_k^2}{\ln t}}^{t} \mathbb{P}\left[\Delta_{s,j} \ge \overline{\Delta_j}\right]$$
$$\le \sum_{s=\frac{t\Delta_k^2}{\ln t}}^{t} \frac{1}{s^3}$$
$$\le \frac{1}{2}\left(\frac{\ln t}{t\Delta_k^2}\right)^2$$
$$\le \frac{1}{t}\frac{\lambda\ln K}{8K^2\beta^2},$$

and the last step follows by definition of $t_{\min}$.

**Control of $\tilde{\Delta}_{t,i}$** We have that:

$$\mathbb{P}\left[\tilde{\Delta}_{t,i} \le \frac{1}{2}t\Delta_i\right] = \mathbb{P}\left[t\Delta_i - \tilde{\Delta}_{t,i} \ge \frac{1}{2}t\Delta_i\right]$$
$$\le \mathbb{P}\left[\left(t\Delta_i - \tilde{\Delta}_{t,i} \ge \frac{1}{2}t\Delta_i\right) \wedge \left(\kappa_t \le \frac{1.25t\Delta_i^2}{\beta\ln t}\right) \wedge \left(\nu_t \le \frac{4t^2\Delta_i^2}{\beta\ln t}\right)\right]$$
$$+ \mathbb{P}\left[\left(\kappa_t \ge \frac{1.25t\Delta_i^2}{\beta\ln t}\right) \vee \left(\nu_t \ge \frac{2t^2\Delta_i^2}{\beta\ln t}\right)\right]$$

We set $\nu = \frac{2t^2\Delta_i^2}{\beta\ln t}$, $\kappa = \frac{1.25t\Delta_i^2}{\beta\ln t}$, $\delta = \frac{1}{Kt}$, and we recall that $\beta = 320$. Then:

$$\sqrt{2\nu \ln\left(\frac{1}{\delta}\right)} + \frac{\kappa \ln\left(\frac{1}{\delta}\right)}{3} = \sqrt{2\frac{2t^2\Delta_i^2}{\beta \ln t}\ln(Kt)} + \frac{\frac{1.25t\Delta_i^2}{\beta \ln t}\ln(Kt)}{3}$$

$$\leq \sqrt{8\frac{t^2\Delta_i^2}{\beta \ln t}\ln(t)} + \frac{\frac{1.25t\Delta_i^2}{\beta \ln t}\ln t}{3} \qquad (21)$$

$$\leq t\Delta_i\left(\frac{2\sqrt{2}}{\sqrt{\beta}} + \frac{2.5}{3\beta}\right)$$

$$\leq \frac{1}{2}t\Delta_i,$$

where equation (21) is due to $t \geq t_{\min}(i) \geq K$, so $\ln(Kt) \leq 2\ln t$. We can then use Theorem 6 and get:

$$\mathbb{P}\left[\tilde{\Delta}_{t,i} \leq \frac{1}{2}t\Delta_i\right] \leq \frac{1}{t}\frac{\lambda \ln K}{4K^2\beta^2} + \frac{1}{Kt} = \frac{1}{t}\left(\frac{\lambda \ln K}{4K^2\beta^2} + \frac{1}{K}\right).$$

Using this bound, summing on $t$ gives

$$\sum_{t=t_{\min}(i)}^{T} \mathbb{E}\left[q_{t,i}\right] \leq \sum_{t=t_{\min}(i)}^{T} \mathbb{E}\left[e^{-\frac{1}{2}t\eta_t\Delta_i}\mathbb{1}\left[\tilde{\Delta}_{t,i} \geq \frac{1}{2}t\Delta_i\right] + \mathbb{1}\left[\tilde{\Delta}_{t,i} \leq \frac{1}{2}t\Delta_i\right]\right]$$

$$\leq \sum_{t=t_{\min}(i)}^{T} \left(e^{-\frac{1}{2}t\eta_t\Delta_i} + \frac{1}{t}\left(\frac{\lambda \ln K}{4K^2\beta^2} + \frac{1}{K}\right)\right),$$

which finishes the proof.

### D.3   Control over the Exploration

We now provide a more general version of Proposition 3.

**Proposition 8.** *Let $S_1, S_2, \ldots$ be a sequence of exploration sets generated by playing algorithm 1 the conditions of Lemma 1 with $\gamma = 4$, $\beta = 320$ and $\lambda \in [1, K]$. Then, the contribution of the extra exploration can be bounded as:*

$$\sum_{t=t_{\min}}^{T}\sum_{i:\Delta_i>0} \Delta_i\mathbb{E}\left[\varepsilon_{t,i}\right] \leq \sum_{t=t_{\min}}^{T}\mathbb{E}\left[\sum_{i\in S_t:\Delta_i>0}\frac{4\beta \ln t}{t\Delta_i}\right] + \frac{\lambda \ln T \ln K}{4K\beta^2} + 12K + 3.$$

*Proof.* By definition of $\xi_{t,i}$, we can decompose the contribution of the extra exploration as follows.

$$\sum_{t=t_{\min}}^{T}\mathbb{E}\left[\varepsilon_{t,i}\right]\sum_{i:\Delta_i>0}\Delta_i \leq \sum_{t=t_{\min}}^{T}\sum_{i:\Delta_i>0}\Delta_i\mathbb{E}\left[\min\{1, \xi_{t,i}\}\right]$$

$$\leq \sum_{t=t_{\min}}^{T}\mathbb{E}\left[\sum_{i\in S_t:\Delta_i\geq 0}\Delta_i\mathbb{E}\left[\min\left\{1, \frac{\beta \ln t}{t\hat{\Delta}_{t,i}^2}\right\}\right]\right]$$

$$+ \sum_{t=t_{\min}}^{T}\sum_{i:\Delta_i>0}\Delta_i\frac{4}{t^2},$$

where in the first step we use the $\min$ term to ensure that we have an upper bound on this quantity in the cases where the bounds on $\hat{\Delta}_{t,i}$ do not hold. The last step consists in upper bounding the exploration of arms that are not in $S_t$ by adding $\frac{4}{t^2}$ to all arms, and in counting $\mathbb{E}\left[\frac{\beta \ln t}{t\hat{\Delta}_{t,i}^2}\right]$ only for arms $i$ that are in the exploration set $S_t$.

Thus the second term is bounded as:

$$\sum_{t=t_{\min}}^{T} \sum_{i:\Delta_i>0} \Delta_i \frac{4}{t^2} \leq K \sum_{t=1}^{T} \frac{4}{t^2} \leq 8K. \tag{22}$$

In order to bound the first term, we recall that for $t \geq t_{\min}$, for any $i \in V$:

$$\mathbb{E}\left[\min\left\{1, \frac{\beta \ln t}{t\hat{\Delta}_{t,i}^2}\right\}\right] \leq \mathbb{E}\left[\frac{\beta \ln t}{t\hat{\Delta}_{t,i}^2} \mathbb{1}\left[\hat{\Delta}_{t,i} \geq \frac{1}{2}\overline{\Delta}_i\right] + \mathbb{1}\left[\hat{\Delta}_{t,i} \leq \frac{1}{2}\overline{\Delta}_i\right]\right]$$

$$\leq \frac{4\beta \ln t}{t\Delta_i^2} + \mathbb{P}\left[\hat{\Delta}_{t,i} \leq \frac{1}{2}\overline{\Delta}_i\right]$$

$$\leq \frac{4\beta \ln t}{t\Delta_i^2} + \left(\frac{\ln t}{t\Delta_i^2}\right)^{\gamma-2} + \frac{2}{Kt^{\gamma-1}} + 2\left(\frac{1}{t}\right)^{\frac{\beta}{10}}$$

$$\leq \frac{4\beta \ln t}{t\Delta_i^2} + \frac{1}{t}\frac{\lambda \ln K}{4K^2\beta^2} + \frac{2}{Kt^3} + 2\left(\frac{1}{t}\right)^2,$$

which gives:

$$\sum_{t=t_{\min}}^{T} \mathbb{E}\left[\sum_{i \in S_t:\Delta_i>0} \Delta_i \mathbb{E}\left[\min\left\{1, \frac{\beta \ln t}{t\hat{\Delta}_{t,i}^2}\right\}\right]\right]$$

$$\leq \sum_{t=t_{\min}}^{T} \mathbb{E}\left[\sum_{i \in S_t:\Delta_i>0} \Delta_i \left(\frac{\beta \ln t}{t\Delta_i^2} + \frac{1}{t}\frac{\lambda \ln K}{4K^2\beta^2} + \frac{2}{Kt^3} + 2\left(\frac{1}{t}\right)^2\right)\right]$$

$$\leq \sum_{t=t_{\min}}^{T} \mathbb{E}\left[\sum_{i \in S_t:\Delta_i>0} \frac{4\beta \ln t}{t\Delta_i}\right] + \frac{\lambda \ln T \ln K}{4K\beta^2} + 3 + 4K.$$

$\square$

### D.4 Proof of Theorem 3 and Corollary 1

The proof of Theorem 3 follows from the propositions in this section.

*Proof of Theorem 3.* We want to bound the pseudo-regret of algorithm 1 run with parameters defined in Lemma 1 with $\gamma = 4$, $\beta = 320$ and $\lambda = \alpha$. The pseudo-regret can be decomposed by treating the first $t_{\min}$ rounds like in the adversarial case, and by using a refined bound in the stochastic regime.

$$R_T = R_{t_{\min}} + \sum_{i\,:\,\Delta_i>0} \sum_{t=t_{\min}}^{T} \Delta_i \mathbb{E}[p_{t,i}]$$

$$\leq R_{t_{\min}} + \sum_{i\,:\,\Delta_i>0} \Delta_i \sum_{t=t_{\min}}^{T} \left(\mathbb{E}[q_{t,i}] + \mathbb{E}[\varepsilon_{t,i}]\right), \tag{23}$$

First, we apply the second part of Proposition 6 with $\lambda = \alpha$, and deduce that:

$$R_{t_{\min}} \leq \frac{1019\beta K}{\Delta_{\min}^2} \left(\ln\left(\frac{\beta K}{\Delta_{\min}}\right)\right)^{3/2} + 2K. \tag{24}$$

Then we bound the contribution of exponential weights by applying Proposition 7 with $\lambda = \alpha$ and $\eta_t = \sqrt{\frac{\ln K}{2\sum_{s=K}^{t-1}\theta_s}}$. By definition of $\theta_s$, $\sum_{s=K}^{t-1}\theta_s \leq tK$, so $\eta_t \geq \sqrt{\frac{\ln K}{2tK}}$. This gives:

$$\sum_{t=t_{\min}(i)}^{T}\mathbb{E}\left[q_{t,i}\right] \leq \sum_{t=t_{\min}(i)}^{T}\left(e^{-\frac{1}{2}t\eta_t\Delta_i} + \frac{1}{t}\left(\frac{\alpha\ln K}{4K^2\beta^2} + \frac{1}{K}\right)\right)$$

$$\leq \sum_{t=t_{\min}(i)}^{T}\left(e^{-\Delta_i\sqrt{\frac{\ln K}{8K}}\sqrt{t}} + \frac{1}{t}\left(\frac{\alpha\ln K}{4K^2\beta^2} + \frac{1}{K}\right)\right)$$

$$\leq \frac{16K}{\Delta_i^2} + \ln T\left(\frac{\alpha\ln K}{4K^2\beta^2} + \frac{1}{K}\right),$$

and then:

$$\sum_{i:\Delta_i>0}\Delta_i\sum_{t=t_{\min}}^{T}\mathbb{E}\left[q_{t,i}\right] \leq \sum_{i:\Delta_i>0}\Delta_i\left(\frac{6K}{\Delta_i^2} + \ln T\left(\frac{\alpha\ln K}{4K^2\beta^2} + \frac{1}{K}\right)\right)$$

$$\leq \ln T\left(\frac{\alpha\ln K}{4K\beta^2} + 1\right) + \sum_{i:\Delta_i\geq 0}\frac{16K}{\Delta_i}, \tag{25}$$

where the last step follows from Lemma 5. Furthermore, we bound the contribution of the extra exploration by applying Proposition 8 with $\lambda = \alpha$, which gives:

$$\sum_{i:\Delta_i>0}\Delta_i\sum_{t=t_{\min}(i)}^{T}\mathbb{E}\left[\varepsilon_{t,i}\right] \leq \sum_{t=1}^{T}\mathbb{E}\left[\sum_{i\in S_t:\Delta_i>0}\frac{4\beta\ln t}{t\Delta_i}\right] + \frac{\alpha\ln K}{4K\beta^2}\ln T + 12K + 3$$

$$\leq \max_{Ind\in\mathcal{I}(G)}\left\{\sum_{i\in Ind:\Delta_i>0}\frac{4\beta\ln^2 T}{\Delta_i}\right\} + \frac{\alpha\ln K}{4K\beta^2}\ln T + 12K + 3, \tag{26}$$

where the last step follows from Proposition 1: by definition, for all $t$, $S_t$ is a strongly independent set on $G$, and we can upper bound by taking the maximum over all the strongly independent sets of $G$.

Finally, summing over equations (24), (25) and (26) and noting that $14K+3 \leq \frac{\beta K}{\Delta_{\min}^2}$ and $\frac{2\alpha\ln K}{4K\beta^2}+1 \leq 2\alpha$ finishes the proof. $\square$

The Corollary 1 follows the same structure.

*Proof of Corollary 1.* We decompose the regret following equation (23), where $t_{min}$ is defined using $\lambda = \tilde{\alpha}$.

$R_{t_{min}}$ is bounded the first part of Proposition 5, which gives:

$$R_{t_{min}} \leq \frac{160\beta K}{\Delta_{\min}^2}\sqrt{\frac{\tilde{\alpha}}{\lambda}}\ln\left(\frac{\sqrt{\beta}K}{\Delta_{\min}}\right) + 2K. \tag{27}$$

Then $\sum_{i:\Delta_i>0}\Delta_i\sum_{t=t_{\min}}^{T}\mathbb{E}\left[q_{t,i}\right]$ is bounded by Proposition 7 with $\lambda = \tilde{\alpha}$, and using the derivation leading to (25), which gives:

$$\sum_{i:\Delta_i>0}\Delta_i\sum_{t=t_{\min}}^{T}\mathbb{E}\left[q_{t,i}\right] \leq \ln T\left(\frac{\tilde{\alpha}\ln K}{4K\beta^2} + 1\right) + \sum_{i:\Delta_i\geq 0}\frac{16K}{\Delta_i}, \tag{28}$$

and $\sum_{i:\Delta_i>0}\Delta_i\sum_{t=t_{\min}}^{T}\mathbb{E}\left[\varepsilon_{t,i}\right]$ follows from the derivation leading to (26), which gives:

$$\sum_{i:\Delta_i>0}\Delta_i\sum_{t=t_{\min}(i)}^{T}\mathbb{E}\left[\varepsilon_{t,i}\right] \leq \max_{Ind\in\mathcal{I}(G)}\left\{\sum_{i\in Ind:\Delta_i>0}\frac{4\beta\ln^2 T}{\Delta_i}\right\} + \frac{\tilde{\alpha}\ln K}{4K\beta^2}\ln T + 12K + 3, \tag{29}$$

Finally, summing equations (27), (28) and (29) finishes the proof. $\square$

# E Extension to Graphs that Change over Time

*Proof of Theorem 4.*

**Adversarial Regime** In the adversarial regime, the proof follows the analysis with a fixed feedback graph up to equation (12), which gives:

$$\mathcal{R}_T \leq 2K + 2\sqrt{2\ln K}\, \mathbb{E}\left[\mathbb{E}_t\left[\sqrt{\sum_{t=K+1}^{T}\theta_t}\right]\right] + \sqrt{T\ln K}, \tag{30}$$

for $\lambda = 1$. All that remains is to bound $\sum_{t=K+1}^{T}\theta_t$. For the first part of the bound, we use Lemma 8 and Proposition 4 to deduce that for all $t \geq K + 1$:

$$\theta_t \leq \tilde{\alpha}_t,$$

and using this bound on $\theta$ in equation (30) gives:

$$\mathcal{R}_T \leq 4\sqrt{\ln K \sum_{t=1}^{T}\tilde{\alpha}_t} + 2K. \tag{31}$$

For the second part of the bound, we recall that for all $t \geq K + 1$ the exploration parameter is lower bounded and fulfills $p_{t,i} \geq \varepsilon_{t,i} \geq \frac{4}{t^2} \geq \frac{4}{T^2}$. Thus we can apply Lemma 7 at each round $t \geq K + 1$, which gives:

$$\theta_t = \sum_{i \in V}\frac{p_{t,i}}{P_{t,i}} \leq 8\alpha_t \ln\left(KT\right).$$

using this bound on $\theta$ in equation (30) gives:

$$\mathcal{R}_T \leq 9\sqrt{\ln K}\sqrt{\ln(KT)}\sqrt{\sum_{t=1}^{T}\alpha_t} + 2K. \tag{32}$$

Taking the minimum over equations (31) and (32) finishes the proof.

**Stochastic Regime**

The structure of the proof follows from Theorem 3. We decompose the regret following equation (23), where $t_{min}$ is chosen using $\lambda = 1$.

$$R_T \leq R_{t_{\min}} + \sum_{i\,:\,\Delta_i > 0}\Delta_i \sum_{t=t_{\min}}^{T}\left(\mathbb{E}\left[q_{t,i}\right] + \mathbb{E}\left[\varepsilon_{t,i}\right]\right).$$

$R_{t_{min}}$ is bounded using the same approach as for Corollary 1, but using the time varying version of the bound given in equation (31). We bound $\tilde{\alpha}_t \leq K$ at each round, which gives:

$$R_{t_{min}} \leq \frac{160\beta K^{3/2}}{\Delta_{\min}^2}\ln\left(\frac{\sqrt{\beta}K}{\Delta_{\min}}\right) + 2K. \tag{33}$$

Then the second term is bounded by Proposition 7 with $\lambda = 1$, and using the derivation leading to equation (25), which gives:

$$\sum_{i\,:\,\Delta_i > 0}\Delta_i \sum_{t=t_{\min}}^{T}\mathbb{E}\left[q_{t,i}\right] \leq \ln T\left(\frac{\ln K}{4K\beta^2} + 1\right) + \sum_{i:\Delta_i \geq 0}\frac{16K}{\Delta_i}, \tag{34}$$

The last term follows Proposition 8 with $\lambda = 1$, which gives:

$$\sum_{t=t_{\min}}^{T}\mathbb{E}\left[\sum_{i \in S_t:\Delta_i \geq 0}\Delta_i\mathbb{E}\left[\varepsilon_{t,i}\right]\right] \leq \sum_{t=1}^{T}\mathbb{E}\left[\sum_{i \in S_t:\Delta_i \geq 0}\frac{4\beta\ln t}{t\Delta_i}\right] + \frac{\ln T\ln K}{4K\beta^2} + 12K + 3.$$

We recall that because $\sum_{i \in S_t:\Delta_i \geq 0}\Delta_i\mathbb{E}\left[\varepsilon_{t,i}\right] \leq 1$ for all $t$, we can skip rounds that have the largest upper bound on $S_t$ by upper bounding the contribution of such rounds by 1. Let $\tilde{A}_n$ be the $n^{th}$ largest

element in the set containing the strong independence number of $G_t$, with $t \in [1, T]$. Then we can upper bound the first term in Proposition 8 as:

$$\sum_{t=1}^{T} \mathbb{E} \left[ \sum_{i \in S_t : \Delta_i \geq 0} \frac{4\beta \ln t}{t \Delta_i} \right] \leq \inf_{0 \leq n \leq T} \left\{ \max_{S \subset V : |S| = \tilde{A}_n} \left\{ \sum_{i \in S : \Delta_i > 0} \frac{4\beta \ln^2 T}{\Delta_i} \right\} + n \right\}.$$

This gives

$$\sum_{t=t_{\min}}^{T} \mathbb{E} \left[ \sum_{i \in S_t : \Delta_i \geq 0} \Delta_i \mathbb{E} \left[ \varepsilon_{t,i} \right] \right] \leq \inf_{0 \leq n \leq T} \left\{ \max_{S \subset V : |S| = \tilde{\alpha}_n} \left\{ \sum_{i \in S : \Delta_i > 0} \frac{4\beta \ln^2 T}{\Delta_i} \right\} + n \right\}$$
$$+ \frac{\ln T \ln K}{4K\beta^2} + 12K + 3. \tag{35}$$

We finish the proof by summing on equations (33), (34) and (35).

$$R_T \leq \frac{160\beta K^{3/2}}{\Delta_{\min}^2} \ln \left( \frac{\sqrt{\beta}K}{\Delta_{\min}} \right) + 2K + \max_{S \subset V : |S| = \tilde{\alpha}} \left\{ \sum_{i \in S : \Delta_i > 0} \frac{4\beta \ln^2 T}{\Delta_i} \right\}$$
$$+ \ln T \left( \frac{\ln K}{2K\beta^2} + 1 \right) + \sum_{i : \Delta_i > 0} \frac{16K}{\Delta_i} + 12K + 3$$
$$\leq \inf_{0 \leq n \leq T} \left\{ \max_{S \subset V : |S| = \tilde{\alpha}_n} \left\{ \sum_{i \in S : \Delta_i > 0} \frac{4\beta \ln^2 T}{\Delta_i} \right\} + n \right\}$$
$$+ 2 \ln T + \sum_{i : \Delta_i > 0} \frac{16K}{\Delta_i} + \frac{161\beta K^{3/2}}{\Delta_{\min}^2} \ln \left( \frac{\sqrt{\beta}K}{\Delta_{\min}} \right).$$

$\square$