# OpenReview forum: "A Near-Optimal Best-of-Both-Worlds Algorithm for Online Learning with Feedback Graphs"
_NeurIPS.cc/2022/Conference — NeurIPS 2022 Accept_

### Official Review · Reviewer_jtPY · 2022-07-01

**Rating:** 7
**Confidence:** 4
**Soundness:** 4 excellent
**Presentation:** 3 good
**Contribution:** 3 good

**Summary:**

This paper considers online learning with feedback graphs and present an algorithm achieving (almost) best-of-both-world regret bounds in stochastic and adversarial environments. The core idea is a combination of the EXP3++ algorithm (providing adaptivity between regime) and the EXP3.G algorithm (exploiting feedback graphs) and a novel exploration scheme which exploits the structure of the graph. The results are extended to the time-varying feedback graphs.


**Questions:**

- Pseudo codes Algorithms 1 and 2 are not clear when standing alone: it might be better if it is indicated clearer where Algorithm 2 (and Equations (2)-(3)) is injected to Algorithm 1.

- It is a little bit confusing that in Line 216, the observability probability is said to be lower bounded by  $(\beta \log t) / (t \hat{\Delta}^2 )$ in the stochastic case but then in Equation (4), it suggests that this might not be the case in general. My understanding for this is that in stochastic case, we can tune the parameters such that $(\beta \log t) / (t \hat{\Delta}^2 )$ is indeed the minimum of the 3 terms of (4), is this correct?

- The wording in Theorem 2 makes it seem unnecessarily limited: from proofs in Appendix B, results of Theorem 2 should work with any gamma and beta such that epsilon = 1/2 $\sqrt(\lambda \log K/ (t K^2))$, right? The specific choice of gamma and beta is for the purpose of achieving the bounds in Theorem 3, right? Personal opinion: one theorem covering 2 cases (such as Theorem 4) would be preferable.

- The statement that the complexity of Algorithm 2 is O(K^3) seems a little bit overkilled, should it be O(K^2) instead (given an efficient sorting algorithm O(KlogK)? It seems that this is related to the "minimum weight cover set problem" so it might be interesting to look into the literature to reduce the complexity further.

- A small speculation: Section 6 deals with the time-varying feedback graphs in an informed setting. I wonder if this can be extended even further to the case of uninformed setting, e.g., by bounding with MAS number similarly to Alon, Noga, et al. "Nonstochastic multi-armed bandits with graph-structured feedback."  (2017) or at least in some special cases where the precise graphs are unknown when making decisions but we know that they belong to a certain class such as in Vu et al. Path Planning Problems with Side Observations-When Colonels Play Hide-and-Seek (2020) (note that these references are not cited in the current version).



**Limitations:**

the authors adequately addressed the limitations and potential negative societal impact of their work.

**Strengths And Weaknesses:**

Strengths: The paper is well written and easy to follow. The results are interesting and in my opinion, contribute significantly to the community. While the proposed algorithm is built upon previously known algorithms, the presented exploration scheme is, as far as I know, novel and provides interesting insight.

Weakness: No significant weaknesses. Two minor comments/suggestions: 1) It seems that there is still room to improve the complexity of Algorithm 2; 2) some numerical experiments comparing the proposed algorithm with non-adaptive benchmarks (showing the trade-off of adaptivity and performance) could be interesting.

---

> ### Author Response · Authors · 2022-08-02
> **To Reviewer jtPY, questions 1-3**
>
> The authors would like to thank the reviewers for their careful reading of the paper and feedback. We will address the questions raised.
>
> >  Pseudo codes Algorithms 1 and 2 are not clear when standing alone: it might be better if it is indicated clearer where Algorithm 2 (and Equations (2)-(3)) is injected to Algorithm 1.
>
> We will try to improve the structure of the algorithm. In particular, we will consider explicitly referring to the use of Algorithm 2 in the line of Algorithm 1 where we update $\varepsilon_{t, i}$.
>
> > It is a little bit confusing that in Line 216, the observability probability is said to be lower bounded by  $\frac{\beta \ln t}{t \hat \Delta_{t, i}^2}$ in the stochastic case but then in Equation (4), it suggests that this might not be the case in general. My understanding for this is that in stochastic case, we can tune the parameters such that  $\frac{\beta \ln t}{t \hat \Delta_{t, i}^2}$ is indeed the minimum of the 3 terms of (4), is this correct?
>
>  It is important to note that in equation (4), the third term is the one that decreases the fastest as a function of $t$. So eventually $t$ becomes large enough such that $o_{t, i}$ is equal to $\frac{\beta \ln t}{t \hat \Delta_{t, i}^2}$. The choice of parameter $\beta$, as expressed in Lemma 1 is fairly large, so the first and second part of equation (4) ensure that the exploration does not get so large in the initial rounds that it would affect the guarantees in the adversarial regime.
>
> > The wording in Theorem 2 makes it seem unnecessarily limited: from proofs in Appendix B, results of Theorem 2 should work with any gamma and beta such that epsilon = 1/2 , $\sqrt{\frac{\lambda \log K}{t K^2}}$ right? The specific choice of gamma and beta is for the purpose of achieving the bounds in Theorem 3, right? Personal opinion: one theorem covering 2 cases (such as Theorem 4) would be preferable.
>
> The choice of $\gamma$ and $\beta$ is actually fairly restricted. For one, similarly to the EXP3++ algorithm (Seldin and Lugosi, 2017), we require $\gamma \geq 3$ and $\beta \geq 64(\gamma + 1) \geq 256$ in order to derive high probability lower bounds on the gap estimates $\hat \Delta_{t, i}$. The choice of $\gamma = 4$, which forces to take $\beta \leq 320$, is due to the equation in line 241. That step is key in Theorem 3. If we were to keep the parameterization of $\gamma = 3$ and $\beta = 256$ like in EXP3++, the term $ 2 \tilde \alpha \ln T$ in the bound of Theorem 3 would be replaced by a term of order $\sum_{i: \Delta_i > 0} \frac{\left(\log T\right)^2}{\Delta_i^*}$, which would dominate the regret and give a dependency on all the suboptimal arms of the graphs.
>
> Concerning the choice of separating Theorems 2 and 4. The two algorithms share a lot in common. It can be noted that the strategy of tuning the learning rate according to $\theta$ in Theorem 4 can seamlessly be applied to Theorem 2 as well, which could allow us to achieve bounds that depend on both the independence number and the strong independence number, depending on which approach we use to bound the $\theta$ term. However, in Theorem~2, we use the knowledge of the strong independence number in order to tune $\epsilon$ rather than the independence number. In Theorem 4, as neither of those quantities are known, the exploration is reduced, which comes at the cost of an increased factor $\sqrt K$ in the last term of the regret bound in the stochastic case, which is a constant.  If we argue that this extra dependence in $K$ is not problematic, then Theorem 4 can be preferred in all cases and we could merge the results of Theorem 2 and 4, in order to derive simultaneous results in terms of the independence number and the strong independence number.

---

> ### Author Response · Authors · 2022-08-02
> **To reviewer jtPY, questions 4 and 5**
>
> > The statement that the complexity of Algorithm 2 is $O(K^3)$ seems a little bit overkilled, should it be $O(K^2)$ instead (given an efficient sorting algorithm $O(K \log K)$? It seems that this is related to the "minimum weight cover set problem" so it might be interesting to look into the literature to reduce the complexity further.
>
> The statement of Algorithm 2 being $O(K^3)$ comes from the following analysis. The initial sorting can be done in time $O(K \log K)$, and as the list has $K$ elements, and the operations from the algorithm only remove elements from that list, parsing that list and 'removing' elements from it each take at most $K$ steps. With this simple analysis, we do not take advantage of the list reducing it size over time, and we agree that $O(K^2)$ seems to be a reasonable complexity for the presented algorithm, and will look into the minimum weight cover set problem in order to see how we can improve the complexity of that part of the problem.
>
> > A small speculation: Section 6 deals with the time-varying feedback graphs in an informed setting. I wonder if this can be extended even further to the case of uninformed setting, e.g., by bounding with MAS number similarly to Alon, Noga, et al. "Nonstochastic multi-armed bandits with graph-structured feedback." (2017) or at least in some special cases where the precise graphs are unknown when making decisions but we know that they belong to a certain class such as in Vu et al. Path Planning Problems with Side Observations-When Colonels Play Hide-and-Seek (2020) (note that these references are not cited in the current version).
>
> Thank you for the extra reference, we will check it. We refer to the first paper from Alon et al. (2017) that you mentioned, but it appears that we cited the ArXiv version of the paper rather than the published version, which we will fix. The reason why we initially focused on the informed setting comes from the lower bound in Online Learning with Feedback Graphs without the Graphs (Cohen et al., 2016), which shows that in the adversarial regime , the learner needs to be able to observe the full graph in order to achieve a dependency on the independence number in the regret bound and that it is not sufficient to only observe the neighborhood of the action played. The setting discussed by  Alon et al. (2017) that you refer to gives access to the full feedback graph at the end of each round, which does not contradict the lower bound. Adapting to such a setting, or to the one of the other paper that you mentioned is an interesting task for future work. Algorithm 2 relies on the structure of the graph in order to ensure that all arms are sufficiently explored. We would expect to require more assumptions on the graph structure if we want to continue using Algorithm 2 in such setting. If the graph was allowed to arbitrarily change from one round to another, the structure of Algorithm 2 would not allow us to ensure that all arms are sufficiently explored as Proposition 1 would not hold.

---

### Official Review · Reviewer_5QMT · 2022-07-03

**Rating:** 6
**Confidence:** 4
**Soundness:** 3 good
**Presentation:** 3 good
**Contribution:** 3 good

**Summary:**

This paper considers the problem of achieving best-of-both-worlds guarantees in the bandits with graph feedback. Specifically, the authors propose an algorithm based on a combination of EXP3.G [Alon et al., 2015] designed for classic bandits with graph feedback problem and EXP3++ [Seldin et al., 2017] designed for getting best-of-both-worlds for MAB. In the adversarial environment, they achieves $\tilde{O}(\sqrt{\alpha T})$ near optimal regret and $O(\max_I \sum_{i\in I}(\frac{\ln^2 T}{\Delta_i}))$ regret bound. The algorithm can also be extended to the time-varying feedback graph setting and achieve the corresponding regret bounds in both the adversarial and the stochastic setting.

**Questions:**

- Could the authors explain more about the relationship between the obtained regret bound in the stochastic environment and the instance optimal regret bound in this environment?

**Ethics Review Area:**

["I don’t know"]

**Limitations:**

Yes, the authors addressed the limitations of the work.

**Strengths And Weaknesses:**

Strength:
- The paper is very well written and the designed algorithm is intuitive and well explained.
- This paper improves upon the previous work by [Erez and Koren 2021] in both the adversarial regret bound and stochastic regret bound. Specifically, in the adversarial environment, the obtained regret bound is optimal ignoring logarithmic factors.
- From the algorithm perspective, the algorithm generalizes the EXP3++ algorithm, which works for MAB problem, to graph bandits with a new specific exploration set, which is a maximal independent and dominating set constructed according to the empirical gap.
- I checked the proofs for each theorem in the main text and the appendix and they look correct to me.


Weakness:
- One issue is about the optimality of the regret bound in the stochastic environment. I thought that the instance-optimal regret bound in the stochastic graph bandits is O(c\ln T) with c defined by an optimization problem with respect to the graph structure. I wonder how large the gap is between the instance-optimal regret bound and the bound derived in this paper?

- One minor issue concerns me is the novelty of this paper as the algorithm generalizes the EXP3++ algorithm with a new but intuitive exploration over the nodes. Based on the exploration, it can be shown that each node can be explored with probability $1/t\hat{\Delta}_i^2$ as some node with smaller gap will be chosen in the exploration set according to the construction. For the other derivations, they seem similar to the analysis in [Alon et al., 2015] and [Seldin et al., 2017].

- Another minor issue I do not understand is that why the results presented by the author for the fixed graph setting has $\tilde{\alpha}$ dependence in the adversarial regret bound while this can indeed be improved to $\alpha$ as shown in the appendix with an additional $\ln T$ factor. I think here the improvement from $\tilde{\alpha}$ to $\alpha$ is more important as this could be an improvement of a factor of $K$ for some specific graph.

Typos:
- line 483: $\alpha_s$ -> $a_s$
- line 575-576: The first inequality should be $\leq\frac{1}{2}n\lambda$ instead of $\geq n\lambda$.
- line 632: $1+\ell_{s,i^*}$ -> $1+\tilde{\ell}_{s,i^*}$

---

> ### Author Response · Authors · 2022-08-02
> **To reviewer 5QMT, about weaknesses**
>
> The authors would like to thank the reviewers for their careful reading of the paper and feedback. We will address the questions raised.
>
> > One issue is about the optimality of the regret bound in the stochastic environment. I thought that the instance-optimal regret bound in the stochastic graph bandits is O(c\ln T) with c defined by an optimization problem with respect to the graph structure. I wonder how large the gap is between the instance-optimal regret bound and the bound derived in this paper?
>
> The instance optimal regret bound is indeed in $O(c \ln T)$, where $c$ is the solution of an optimization problem. It is an interesting direction whether it is possible to recover this types of bounds in a best-of-both-worlds regime.
>
> > One minor issue concerns me is the novelty of this paper as the algorithm generalizes the EXP3++ algorithm with a new but intuitive exploration over the nodes. Based on the exploration, it can be shown that each node can be explored with probability $\frac{1}{t \hat \Delta_{i}^2}$ as some node with smaller gap will be chosen in the exploration set according to the construction. For the other derivations, they seem similar to the analysis in [Alon et al., 2015] and [Seldin et al., 2017].
>
> It can be noted that our analysis improves upon the bounds of Seldin and Lugosi (2017) in the case of bandit feedback (see our answer to reviewer TM6a).
>
> > Another minor issue I do not understand is that why the results presented by the author for the fixed graph setting has $\tilde \alpha$ dependence in the adversarial regret bound while this can indeed be improved to $\alpha$ as shown in the appendix with an additional  $\ln T$ factor. I think here the improvement from $\tilde \alpha$ to $\alpha$ is more important as this could be an improvement of a factor of $K$ for some specific graph.
>
> Choosing whether to present the results of Theorems 2 and 3 in terms of $\alpha$ or $\tilde \alpha$ was discussed at the time of writing the paper. The results that you mentioned indeed hold in terms of $\alpha$ at the cost of a $\log T$ factor by simply changing $\tilde \alpha$ into $\alpha$ in the learning rate and the exploration parameter. The choice was made to postpone the analysis of the case depending on $\alpha$ to Theorem 4, as the choice of tuning the learning rate in terms of $\theta$ led to a better dependency in $\sqrt{\log T}$ instead of $\log T$. Nothing in the analysis prevents us to use the learning rate of Theorem 4 in Theorems 2 and 3, and we will rewrite the bounds, such that both the results in terms of $\tilde \alpha$ and $\alpha$ hold simultaneously.
>
> > Typos
>
> Thank you for pointing them out, we will to fix them.

---

> > ### Comment · Reviewer_5QMT · 2022-08-09
> > **Thanks for the response**
> >
> > Thanks for the authors response and my issues are addressed by the authors. Although the current regret bound in the stochastic world may not match the instance-optimal regret bound, the algorithm and the bounds are interesting based on my understanding.

---

> ### Author Response · Authors · 2022-08-02
> **To reviewer 5QMT, about questions**
>
> > Could the authors explain more about the relationship between the obtained regret bound in the stochastic environment and the instance optimal regret bound in this environment?
>
> One instance where the difference between the problem dependent $c$ and the independence number can be the large can be described as follow:
> Consider an undirected graph with $K$ arms that has a star structure, where there is one arm in the middle of the star and has a large suboptimality gap of $\Delta_max$. Then $K-2$ arms have an edge with the center of the star, and each of them have a smaller suboptimality gap $\Delta_min$. Finally, the optimal arm, which has suboptimality gap $0$, is independent of the rest and does not contain any edge that connects it to the rest of the graph.
>
> We recall that in order to identify the best arm, in a stochastic problem, each suboptimal arm $i$ has to be observed at a rate of at least $\frac{1}{t\Delta_i^2}$. In this instance of the problem, we assume that $\Delta_{max} > \Delta_{min} > 0$. Our greedy strategy would construct an exploration set by adding all arms besides for the one in the center of the star.
> These arms form the largest independence set of the graph. Each of the $K-2$ arms on a leaf of the star needs to be explored, hence played, at rate of at least $\frac{1}{t\Delta_{min}^2}$, and each time we do so $\Delta_{min}$ is added to the pseudo-regret.
> In total, the cost of exploration for this strategy affects the regret by a factor:
>     $R_T \geq (K-2)\frac{ \Delta_{min}}{\Delta_{min}^2}\log T = \frac{(K-2)}{\Delta_{min}} \log T. $
>  On the opposite side, if the learner leverages his knowledge of the graph structure, it can then become more interesting to play the arm in the center of the star, as it allows to explore all the $K-2$ leaves simultaneously. Each leaf still needs to be observed at rate at least $\frac{1}{t\Delta_{min}^2}$, and playing the arm in the middle costs $\Delta_{max}$ at each round.
> Doing so results in contributing to the pseudo-regret by a factor of at least:
> $ R_T \geq \frac{ \Delta_{max}}{\Delta_{min}^2}\log T.$
> The second approach becomes more interesting when the gaps $\Delta_{min}$ and $\Delta_{max}$ are close, meaning that our approach can be suboptimal by up to a factor of $\tilde \alpha$.
>
> It is an interesting direction for future work to investigate how to leverage the graph knowledge to achieve bounds that take advantage of the graph structure, in particular in the case of directed graphs.

---

### Official Review · Reviewer_TM6a · 2022-07-09

**Rating:** 7
**Confidence:** 3
**Soundness:** 4 excellent
**Presentation:** 4 excellent
**Contribution:** 3 good

**Summary:**

This paper studies online learning in a new setting of directed graph feedback. By combining the ideas of EXP3++ and EXP3.G, they suggest an algorithm, EXP3.G++ that can achieve near-optimal regret bound in both stochastic and oblivious adversarial settings.

**Questions:**

Can the authors commend on the performance of EXP3.G++ in extreme cases of bandit and full information?

Can the authors commend the applications of graph feedback (especially directed graph feedback) in real-world situations?


**Limitations:**

There is no potential negative societal impact.

**Strengths And Weaknesses:**

I enjoyed reading the paper and all the results are sound and well presented. The regret bound for both adversary and stochastic settings improves the previous results (Erez and Koren (2021)) by log(T) factors and achieves a near lower bound guarantee. Further, the authors also provide an extension for changing the feedback graphs and provide a regret bound for EXP3.G++ in this case.

---

> ### Author Response · Authors · 2022-08-02
> **To Reviewer TM6a**
>
> The authors would like to thank the reviewers for their careful reading of the paper and feedback. We will address the questions raised.
>
>
> > Can the authors commend on the performance of EXP3.G++ in extreme cases of bandit and full information?
>
> In the case of a pure bandit feedback, it is sufficient to look at the bounds using $K$ as the independence number. In the adversarial regime, seen in Theorem 2, we achieve the exact same bound as EXP3++.
> In the stochastic regime, the results of Theorem 3 differ slightly from the results of EXP3++. On the positive side we refined the analysis of the initial rounds, which improved the additive term from $ O\left( \sum_{i: \Delta_i > 0} \frac{K}{\Delta_i^3}\right)$ to $ O\left(\frac{K}{\Delta_{min}^2}\right)$.
> On the negative side, the choice of $\gamma$ and $\beta$ of EXP3++.G are suboptimal if we only consider a bandit feedback, but it only increases the stochastic bound by a small multiplicative constant of $5/4$.
>
> In the case of the full information regime, one may use the Hedge algorithm as suggested by Mourtada and Gaiffas (2018) as it is best-of-both worlds optimal. Thus, we did not analyse the algorithm specifically in this regime.
>
> > Can the authors commend the applications of graph feedback (especially directed graph feedback) in real-world situations?
>
> Feedback graphs can be useful in many different settings, in particular for problems like digital pricing or auctions, where there is a structure. For example, if the different actions of your game correspond to different prices, you can use feedback graphs to model that if a costumer buys an item at a certain price, he would also have bought it at any cheaper price, and thus give feedback not only for the arm played but to the arms corresponding to cheaper prices.

---

### Official Review · Reviewer_VD7D · 2022-07-13

**Rating:** 7
**Confidence:** 3
**Soundness:** 4 excellent
**Presentation:** 3 good
**Contribution:** 3 good

**Summary:**

This paper studies the multi-armed bandit problem with graph feedbacks where the learner would receive the loss information of all the arms $j$ that $e(i,j) \in G$ after selecting arm $i$, where $G$ here is the feedback graph. The proposed algorithm aims at achieving the best of both worlds guarantee, that is, achieving $\widetilde{O}(\log T)$ regret with stochastic losses while ensuring worst-case robustness $\widetilde{O}(\sqrt{\alpha T})$ where $\alpha$ here is s the independence number of the feedback graph.

The idea of the proposed algorithm originates from the EXP3++ algorithm for stochastic and adversarial bandits, which achieves the near-optimal best of both worlds result for normal multi-armed bandits. To extend the EXP3++ algorithm to the graph feedback setting, the learner takes the idea of EXP3.G which was designed only for adversarial graph feedback, and build up a novel exploration scheme to carefully estimate the gaps, in order to control the magnitude of exploitation and exploration.



**Questions:**

1. Though the $\frac{1}{\Delta^2}$ term in Theorem 3 seems inevitable (as it also appears in EXP3++), is it possible to drop this term as the some best of both worlds results of MAB (including BROAD of Wei and Luo (2018) and Tsallis-INF of Zimmert and Seldin (2019)) do not have this term.

2. Is it possible that SAPO approach could achieve better regret bounds?

**Limitations:**

None.

**Strengths And Weaknesses:**

The paper shows that the best of both worlds guarantee can be achieved by EXP3++ approach for weakly observable feedback graphs. Moreover, it seems that the algorithm can achieve $O(\sqrt{\alpha T})$ regret for adversarial losses while having more logarithmic terms for the stochastic setting. The specific contribution is the new exploration scheme and may be applied in other questions.

This paper does not have any specific weakness. The writing is clean and well-organized.

---

> ### Author Response · Authors · 2022-08-02
> **To reviewer VD7D**
>
> The authors would like to thank the reviewers for their careful reading of the paper and feedback. We will address the questions raised.
>
> > Though the $\frac{1}{\Delta^2}$ term in Theorem 3 seems inevitable (as it also appears in EXP3++), is it possible to drop this term as the some best of both worlds results of MAB (including BROAD of Wei and Luo (2018) and Tsallis-INF of Zimmert and Seldin (2019)) do not have this term.
>
> Note that the term $\frac{1}{\Delta^2}$ has improved compared to the EXP3++ analysis, that contains a $\frac{1}{\Delta^3}$ term. It is still an open question whether it would be possible to get completely rid of this term.
> This being said, Erez and Koren (2021) noted that it was not trivial to achieve near best-of-both worlds guarantees using a FTRL style of algorithm. In particular, it is unclear whether it is possible to achieve algorithms that depend on the independence number of the graph when the regularisation is different from the Shannon entropy (which is used in EXP3), so even if it is not possible to improve upon the analysis of EXP3++ further, there may still be a trade-off between achieving better dependencies on the graph structure and better dependencies on sub-optimality gaps.
>
> > Is it possible that SAPO approach could achieve better regret bounds?
>
> We did not consider the SAPO approach during our analysis. At the moment we do not see how to analyse SAPO in the feedback graph setting. It is also worth noting that the SAPO algorithm is based on an irreversible switch between two operation modes and requires knowledge of the time horizon. If we were considering the adversarial or the stochastic regime separately, then there are tools, such as the doubling trick, that can handle arbitrary time horizon at the cost of a constant. However, in a best-of-both worlds setting, the doubling trick leads to an extra $\log T$ factor in the stochastic regime, due to the number of times the algorithm restarts (see Besson and Kaufmann, "What doubling tricks can and can’t do for multi-armed bandits"). As our approach is only suboptimal by a factor $\log T$ in the stochastic regime (and the additive $\frac{1}{\Delta^2}$), even if a SAPO approach could achieve a best-of-both worlds optimal result for a fixed time horizon, when generalizing to an arbitrary time horizon the gain would not be substantial. It is also worth noting that for the classic MAB setting, the SAPO algorithm is suboptimal by a $\sqrt{\log (T)}$ factor in the adversarial regime, which further limits the potential of this approach being competitive.

---

### Meta-Review · Area_Chair_jPY6 · 2022-08-24

**Recommendation:** Accept
**Confidence:** Certain

**Metareview:**

The reviewers came to consensus that this paper makes a good progress on the online learning with feedback graphs. I agree with these opinions and please polish the manuscript so that the minor concerns raised by the reviewers become clear in the final version.

**Award:**

No

---

### Decision · Program_Chairs · 2022-09-14

Accept